

# Characteristics and fate of isolated permafrost patches in coastal Labrador, Canada

Robert G. Way[1,2], Antoni G. Lewkowicz[1], Yu Zhang[3]

[1]Department of Geography, Environment and Geomatics, University of Ottawa, Ottawa, K1N6N5, Canada
[2]Labrador Institute, Memorial University of Newfoundland, Happy Valley-Goose Bay, A0P1E0, Canada
[3]Canada Centre for Mapping and Earth Observation, Natural Resources Canada, Ottawa, K1A0E4, Canada

*Correspondence to*: Robert G. Way (rway024@uottawa.ca)

**Abstract.** Bodies of peatland permafrost were examined at five sites along a 300 km transect spanning the isolated patches permafrost zone in the coastal barrens of southeastern Labrador. Mean annual air temperatures ranged from +1°C in the south (latitude 51.4°N) to -1.1°C in the north (53.7°N) while mean ground temperatures at the top of permafrost varied respectively from -0.7°C to -2.3°C with shallow active layers (40-60 cm) throughout. Small surface offsets due to wind scouring of snow from the crests of palsas and peat plateaux, and large thermal offsets due to thick peat are critical to permafrost, which is therefore absent in wetland, forested and forest-tundra areas inland, notwithstanding average air temperatures much lower than near the coast. Most permafrost peatland bodies are less than 5 m thick with a maximum of 10 m with steep geothermal gradients. One-dimensional thermal modelling for two sites showed that they are in equilibrium with the current climate, but the permafrost mounds are generally relict and could not form today without the low snow depths that result from a heaved peat surface. Despite the warm permafrost, model predictions using downscaled global warming scenarios (RCP2.6, 4.5 and 8.5) indicate that perennially frozen ground will thaw from the base up and may persist at the southern site until the middle of the 21[st] Century. At the northern site, permafrost is more resilient, persisting to the 2060s under RCP8.5, the 2090s under RCP4.5, or beyond the 21[st] century under RCP2.6. Despite evidence of peatland permafrost degradation in the study region, the local-scale modelling suggests that the southern boundary of permafrost may not move as quickly as had previously been thought.

**Key words:** palsa, peat plateau, electrical resistivity tomography, ground thermal modelling



## 1 Introduction

Global and regional models project that the southern boundary of permafrost will shift northwards over the next century as the climate warms (e.g. Koven et al., 2013; Zhang et al., 2008a, 2008b). In the zone of isolated patches, permafrost underlies less than 10% of the landscape and is mainly present in frozen peatlands (Smith and Riseborough, 2002). Peatland permafrost is considered to be vulnerable to thaw in response to projected regional warming, modifying ecosystem properties and changing landscape development processes (Tarnocai, 2006, 2009). Projected loss of peatland permafrost is also expected to result in an increase in atmospheric carbon (Schuur et al., 2015) although possibly less than previously anticipated (Cooper et al., 2017; Turetsky et al., 2007). To date, most field-based studies of changes in peatland permafrost have focused on the sporadic and extensive discontinuous zones (e.g. Halsey et al., 1995; James et al., 2013; Kwong and Gan, 1994; Sjöberg et al., 2015; Swindles et al., 2016; Thibault and Payette, 2009). A better understanding of the current distribution and characteristics of isolated bodies of permafrost is needed to assess the likelihood of their persistence or degradation in the face of climate warming.

Peatland permafrost features include palsas and peat plateaux, both resulting from ice segregation processes (Gurney, 2001; Seppälä, 1982, 2011). Palsas form in an otherwise unfrozen peatland due to localized snow redistribution caused by wind-scouring of micro-topographic highs which allows greater ground heat loss in winter, deeper frost penetration and surface heaving due to segregated ice formation (Seppälä, 1982, 2011). The elevated ground surface promotes drying of near-surface peat in the summer, reducing its thermal conductivity and diminishing summer heat transfer into the ground (Allard and Rousseau, 1999; An and Allard, 1995; Seppälä, 2011). Peat plateaux form due to similar processes operating where extensive areas of peatlands are well-drained, allowing for larger areas of frost heave, wind-scouring in the winter and less heat penetration in the summer (Zoltai, 1972; Zoltai and Tarnocai, 1975).

Peatland permafrost bodies have partially or completely thawed at many locations within the Circumpolar North over the past few decades, including Scandinavia (Borge et al., 2017), eastern Canada (Thibault and Payette, 2009) and western Canada (e.g. Halsey et al., 1995; Quinton et al., 2011). Despite this apparent trend towards degradation, permafrost bodies, first described 35-75 years ago (Brown, 1979, 1975; Dionne, 1984; Hustich, 1939), have persisted at coastal locations in the isolated patches permafrost zone in eastern Canada (Dionne, 1984; Dionne and Richard, 2006). At the same time, regional-scale modelling suggests that considerable permafrost thaw could have occurred in this region over the past 50 years (Way and Lewkowicz, 2016), coinciding with a period of rapidly increasing regional air temperatures (Way and Viau, 2015).

Here we assess the state of peatland permafrost at five locations across the isolated patches zone of southeastern Labrador and easternmost Québec. Permafrost-cored mounds in these wetlands have cultural significance as locations for storing komatiks (towing sleds), berry picking (cloudberries) and cooling meat beneath the surface peat layers (Patricia Way, *personal communication*). The elevated surfaces have also been regularly used by indigenous and settler trappers as locations to establish fox traps (Gary Bird, *personal communication*). We describe field measurements of air temperature,





ground temperature and the spatial extent of permafrost patches using frost probing and electrical resistivity tomography (ERT), and the results of numerical modelling to assess the fate of these isolated bodies of perennially frozen ground as the climate warms.

## 2. Study area

The study area comprises the coastal barrens ecozone in Labrador, from near Rigolet (54.2°N) southwards and extending across the Québec border as far as Blanc Sablon (51.4°N) (Fig. 1a). Regional vegetation cover in this ecozone is sparse forest-and-shrub-tundra on exposed outer coasts but includes dense Black Spruce, Tamarack and White Spruce forests in sheltered areas and farther inland (Roberts et al., 2006). Peat deposits and wetlands are present throughout the region with plateau string bogs clustered in the interior and raised bogs in the coastal zone (Foster et al., 1988; Foster and Glaser, 1986; Glaser, 1992). Surficial materials consist primarily of exposed rock and bedrock covered by glacial tills on outer coasts, and glacial till with scattered glaciofluvial deposits farther inland (Fig. 1b) (Fulton, 1995). Marine and glaciomarine sediments, deposited during Holocene high stands, are present near Blanc Sablon and in areas north of Cartwright.

Mean annual air temperature (2013-2016) along the coast declines northwards from +1.0°C near Blanc Sablon to -0.5°C north of Cartwright, (Fig. 1a) (Way et al., 2017). Lower temperatures are experienced inland in high elevation plateaus and mountainous zones, including the Akami-Uapishk$^U$-KakKasuak-Mealy Mountains National Park Reserve. Air temperatures in the region increased by 1.8°C between 1917 and 2016 (Way and Viau, 2015) but annual temperatures were within ±0.1°C of the long-term means (1948-2016) during the period of field observations (2013-2017). Mean cold-season (December to April [1981-2010]) snow depth averages 109 cm at Cartwright but only 24 cm at Blanc Sablon (Environment and Climate Change Canada, 2017).

The Permafrost Map of Canada (Heginbottom et al., 1995) shows most of the study transect as isolated patches of permafrost, with the extreme southern boundary of the zone just north of Blanc Sablon and small areas at the northeast end of the transect within the sporadic discontinuous zone (Fig. 1b). Palsas and other periglacial features have been described at a reconnaissance level in early work and more recently (e.g. Brown, 1979, 1975; Dionne, 1984; Dionne and Gérardin, 1988; Dionne and Richard, 2006; Hustich, 1939; Roberts et al., 2006), but systematic measurement or analysis of peatland permafrost has been undertaken only near Blanc Sablon (Dionne, 1984; Dionne and Richard, 2006).

## 3. Methods

Climate monitoring stations were established at five peatland permafrost sites (Fig. 1a; Table 1). Additional climate information is available from several non-peatland monitoring stations in the coastal barrens ecozone and farther inland (Fig. 1a). Shielded air temperature at 190-235 cm (because of deep regional snow covers), ground surface temperature (~2-3 cm beneath the surface) and shallow ground temperature (50-120 cm depths) were measured bi-hourly using Onset Hobo Pro V2 (accuracy ± 0.2°C) and/or Maxim Integrated high-resolution ibutton loggers (accuracy ± 0.5°C). Corrections for differing



sensor types are described in Way and Lewkowicz (accepted). The deepest ground temperature measurement was at the depth of probe penetration in mid to late-summer and was considered to approximate the temperature at the base of the annual freeze-thaw layer. Snow depth was measured using vertically arranged low-resolution ibutton loggers (e.g. Lewkowicz, 2008) installed at various heights (typically 10, 20, 30, 40, 50, 60, 80, 100, 140 and 180 cm). Ibutton data were

used to calculate daily snow heights using an empirical threshold established from correlations between the daily temperature ranges at various heights (see Way and Lewkowicz, accepted).

Five shallow boreholes (Table 2) were drilled using the water-jet method and were cased with 1-inch PVC pipe. The base of the permafrost was not reached at two locations (WJD01 and WJD05) due to limited water supply. Ground temperatures at four to six depths within each borehole were recorded bi-hourly with Onset Hobo V2 loggers. Temperatures

at WJD01 were measured with four high resolution ibutton loggers. Ground temperatures for four of the five boreholes are reported for the second and third full years of record to eliminate any thermal influence of water-jet drilling. At WJD05, data are provided only for the second year because this borehole was drilled one year after the others. Short data gaps in the borehole temperature records were infilled using cross-correlation between borehole depths and/or adjacent climate monitoring stations established at each location.

ERT surveys at the study sites (Table 1) were undertaken with an ABEM Terrameter LS profiling system with electrodes arranged in a Wenner configuration. ERT has been widely used for detecting permafrost bodies and estimating permafrost thickness in Canada (Briggs et al., 2016; Douglas et al., 2016; Lewkowicz et al., 2011; Minsley et al., 2016; Way and Lewkowicz, 2015), Scandinavia (Kasprzak, 2015; Sjöberg et al., 2015), the European Alps (Hauck, 2013) and the Tibet Plateau (You et al., 2013). Minimum electrode spacing was 0.5, 1 or 2 m over standard profile lengths of 40-160 m, or

longer where roll-along surveys were performed, giving maximum penetration depths of approximately 6 m, 12 m or 25 m, respectively. RES2DINV software was used to invert the measured resistivities (Loke et al., 2003; Loke and Barker, 1996) with the robust inversion method. Inversion proceeded until the fifth iteration or until the RMS error dropped below 5%, whichever came first. Prior to inversion, ERT profiles were topographically corrected using a handheld GPS (Garmin Oregon 450t) to obtain start point elevations and a Brunton compass to obtain relative elevations. Modelled resistivities are

presented as model blocks with low sensitivity areas (< 0.1) faded to reflect the uncertainties of these sections in the modelled profile. Model blocks are larger near the ends of a profile than beneath its centre, reducing the detail in these portions. The frost table was probed at 1 or 2 m intervals along each ERT transect using a 120 cm titanium probe. Surface vegetation cover was also described in the field, and aerial images were acquired using a DJI Phantom 3 Professional quadracopter.

One-dimensional numerical modelling was undertaken for borehole WJD02 at Cartwright and borehole WJD03 at Blanc Sablon using the finite-difference Northern Ecosystem Soil Temperature model (NEST) (Zhang, 2003). NEST has been used to model change in peatland permafrost features in northern Manitoba (Zhang, 2013) and northern Ontario (Ou et al., 2016a, 2016b). Soil profile properties (e.g. peat thickness) were estimated from observations during water-jet drilling and depth to bedrock was based on the ERT survey results (Table S1). The base of the model was set to a depth of 120 m.



Climate inputs from 1901 to the start of station observations comprised interpolated daily temperature data (10 km resolution) from the Canadian Forest Service (Hutchinson et al., 2009). The year 1900 was infilled with the 1901 monthly data for model initialization purposes. Homogenized station data (Vincent et al., 2012) was used for the remainder of the modelling period, from 1938-2016 for Cartwright, and 1976-2016 for Blanc Sablon (Environment and Climate Change Canada 2017).

The model was calibrated using daily borehole temperatures for 2014-2016. A snow wind-scouring factor (fraction of snowfall blown away from the site) was adjusted until modelled near-surface soil layer temperatures were comparable to observations (Table S1). The parameters were kept constant for future predictions whose climatic inputs were multi-model means derived from ensemble projections for RCP 2.6, 4.5 and 8.5 for 2016-2100 (van Vuuren et al., 2011). The geothermal heat flux was tuned by comparing modelled and observed permafrost thickness and deep ground temperatures. To examine the major controls on ground temperatures, the model was also run for the Blanc Sablon borehole assuming no snow scouring and no peat cover.

## 3 Results

### 3.1 Field observations

Field results are presented from south to north for the five peatland permafrost sites and are summarized in Table 3 and Table S2.

### 3.1.1 Blanc Sablon, QC

A group of five mounds, 0.4-1.4 m high and up to ~16 m in diameter, were investigated at a wetland site located 3.5 km north of Blanc Sablon (Fig. 2a). The mounds were mostly lichen-covered with a few herbs, but some had bare peat surfaces and evidence of wind abrasion and surface cracking. Grasses and sedges were the dominant ground cover in depressions between the mounds and shallow standing water was present at some locations. Shrubs 10-20 cm high, including blueberry and Labrador tea, were present on the sides of the mounds and in the adjacent terrain. Average air temperature (2015-2017) was +0.5°C and snow depths on top of the mound reached 15 cm in late-winter. Ground temperature at the borehole averaged +1.5°C at the surface and -0.7°C at 1 m depth (Fig. 3c). Temperatures were positive at the bottom of the borehole and the permafrost base was estimated to be at a depth of between 3.5 and 4.2 m. The median frost table depth on the elevated mounds was 63 cm (September 11, 2015) while full frost probe penetration (120 cm) was possible in depressions and on all portions of the ERT transects without mounds (Fig. 3). At several of these locations, however, probing detected 15-30 cm of frozen ground approximately 40 cm below the surface that could be penetrated with effort, and which was underlain by unfrozen soil to 120 cm depth.

ERT profile 1 crossed four mound crests while Profile 2 ran perpendicularly across a mound complex that included the climate monitoring station and the borehole (Fig. 2a; Fig. 3d). Both ERT surveys showed positive resistivity anomalies





beneath mounds that end at the edges of the adjacent depressions (Fig. 3a,b). Resistivities >600 Ω.m extended to depths of 3.4 m (P1), 4.5 m (P2) and 4.3 m (P4) on profile 1, and to the base of the survey below P3 on both profiles 1 and 2. Resistivities <400 Ω.m were typical for the near-surface layer and in depressions between each mound as well as at depth beneath P1 and P4.

The field observations and the borehole temperatures lead to the conclusion that the mounds are palsas. The resistivity boundary between frozen and unfrozen ground was inferred to be 500-600 Ω.m, giving permafrost thicknesses of 2.8 m (P1), 3.9 m (P2) and 3.7 m (P4) (Table 3). Permafrost thickness beneath P3 was more challenging to interpret as profile 1 indicated high resistivity values extending to the base of the profile whereas co-located ground temperatures showed unfrozen ground at 4.25 m. The higher resolution profile 2 showed low-sensitivity values (Fig. 3b) beneath the

borehole but high resistivity values nearby that suggest frozen ground may be deeper away from the borehole. Considering the accuracy of the ground temperature loggers (± 0.2°C) and uncertainty inherent to the ERT, the maximum permafrost thickness beneath P3 was estimated to be about 6 m.

The thin layer of frozen ground found at some locations without mounds was interpreted to be a remnant of seasonal frost, but given the September survey date, it may have persisted throughout the thaw season, thereby becoming

permafrost.

Ground temperatures averaged 1.0°C (35 cm depth) and 0.6°C (60 cm depth), at two upland tundra locations 250 m northwest of WJD03 and AMET13 (Table 1). Despite signs of surface wind abrasion, suggesting low snow cover in winter, and the presence of patterned ground in mineral soil (sorted circles and steps), these data indicate that permafrost was absent.

### 3.1.2 Red Bay, NL

A wetland with low mounds, about 2 km north of Red Bay and adjacent to the Trans Labrador Highway, was instrumented with a climate monitoring station and borehole. The mean air temperature was +0.8°C (2015-2017), late-winter snow depth was 25 cm, and ground temperatures averaged +1.7°C (near-surface), -0.7°C at 1 m, -0.01°C at 3 m and +0.3°C at 4.2 m depth (Fig. 4b). Permafrost was therefore present at the borehole between depths of ~0.5-3.5 m .

Two ERT profiles were undertaken. Profile 3 traversed four mounds, 0.3-1 m high, with lichen and herb cover,

shrubs up to 25 cm high on the sides of the mounds and some exposed peat on the crests, terminating in an adjacent wetland (Fig. 2b; Fig. 4).  Sedges and grasses were present in wetter portions at the beginning and end of the profile. Frost tables were encountered continuously beneath the elevated surface of the mounds (median thaw depth: 54 cm on September 12, 2015) whereas full probe penetration was possible in most other areas (Fig. 4a). Exceptions to this occurred from 1-14 m and 68-80 m along the profile where a frozen soil layer (median depth: 60 cm) up to 40 cm thick and overlying unfrozen ground

was present. Surprisingly, this thin frozen layer was encountered even in vegetated areas covered by shallow standing water.

Profile 4 was perpendicular to profile 3 on the opposite side of the main wetland depression, beginning among mounds in the depressed wetland and ending upslope in drier peat-covered terrain (Fig. 5). Peat and lichen cover dominated on the mounds, and low grasses and sedges with some low shrubs in the wetland. Probing revealed frozen ground beneath



the mounds (median thaw depth: 63 cm) but not in the areas between (Fig. 5a). Frost tables were also encountered continuously along the gently sloping peat-covered area (median thaw depth 64 cm).

Profile 3 exhibited high resistivity anomalies (>600 Ω.m) centred beneath four mound crests, with the highest values (>4000 Ω.m) beneath P7 (Fig. 4a). The positive anomalies extended to depths of 2-4 m (Fig. 4a) (Table 3; Table S2)
and were generally underlain by resistivities less than 200 Ω.m. However, higher resistivities extended to the base of the profile beneath P6 and there was a continuous high resistivity layer at the base of the profile, albeit with low sensitivity in the model. Profile 4 also showed positive resistivity anomalies beneath three low mounds, to depths of 3-4 m beneath P9 and P10, and to the base of the profile at P11, at locations where a frost table was encountered. A high resistivity body (>1000 Ω.m) overlain by lower surface values was also evident beneath the second half of the profile and extended to the base where
it joined a layer with high apparent resistivities but low sensitivity in the model.

The field observations and temperature data demonstrate the presence of patchy peatland permafrost at the Red Bay site. The mounds are palsas and the boundary resistivity between frozen and unfrozen ground was between 500-600 Ω.m, giving inferred permafrost thicknesses of 0.8-4.6 m on the two profiles (Table 3; Table S2). The base of permafrost could not be identified beneath P6, nor beneath the peat-covered slope at 38-80 m on Profile 4. However, given the distinctly lower
resistivities in the unfrozen surface layer, the measured frost table depths, the peaty soil, and the presence of a surface bedrock exposure upslope of the end of the profile, it is inferred that permafrost was present and that it extended to the bedrock in this part of profile 4.

### 3.1.3 Cartwright, NL

Three raised bogs containing low peat mounds were investigated in and around the community of Cartwright
(Tables 3; Table S2). The average air temperature was -0.2°C (2015-2017), the near-surface temperature on top of one mound was +1.2°C and late-winter snow depths on the mound crest were 25 cm. Ground temperatures in borehole WJD02 on top of a mound about 100 m from the climate station averaged -1.7°C at 0.5 m, and then warmed with depth to -1.4°C at 1.0 m and +0.3°C at 4.25 m, indicating a permafrost base at approximately 3.6 m, as also observed during water-jet drilling. Temperatures in a shallower borehole (WJD01) about 70 m to the southeast were similar down to its base at 2.15 m. A total
of seven ERT profiles were undertaken in the three bogs (Table 3, profiles 5-11). We present results for two long surveys (profiles 5 and 7) (Fig. 6), and a third shorter survey (profile 9) that traverses borehole WJD02 (Fig. 7).

Profile 5 traversed numerous small elevated peaty areas and six larger peat mounds ranging in height from 0.3-1.3 m (Fig. 6c). The surface cover was primarily lichens and exposed peat in drier areas and low-to-medium height shrubs in wetter zones. Shrubs were taller towards the end of the 240 m long survey and transitioned to White Spruce forest at the
edge of the bog. Frost tables were measurable beneath elevated peat mounds and at some locations with slightly elevated dry peat and the median thaw depth in these areas was 45 cm (July 22, 2014) (Fig. 6a). Full probe penetration occurred wherever the peat was moist or standing water was present. Modelled resistivities ranged from less than 10 Ω.m in the near-surface at the start of the profile to greater than 6000 Ω.m at depths below 10-12 m (although most of these values were in low



sensitivity parts of the model). Positive resistivity anomalies were evident beneath six peat mounds (P12-P17) with values between 600 Ω.m and 4000 Ω.m (Fig. 6a). Resistivities of 200 Ω.m or less were present at greater depths beneath the mounds and in adjacent areas across all but the last 50 m of the profile. Resistivities > 4000 Ω.m extending from the near-surface to the base of the profile between 222 m and 240 m and across almost the whole profile at a depth of 8-10 m, were

interpreted as bedrock based on the local geomorphology (Fig. 6a).

Profile 9 was a detailed survey of two mounds, 0.3-0.6 m high, across borehole WJD02 in the same bog as profile 5. Ground cover was similar, with exposed peat and lichens on the mound and low shrubs and mosses in the wet depressions. Frost tables were measurable at all but three spots along the profile with a median depth of 43 cm (July 24, 2014) (Fig. 7a). Modelled resistivities exceeded 1000 Ω.m to depths of 2 m beneath P24 and 3.5 m beneath P25, with a continuous lower

resistivity layer at the ground surface. A sharp decline in resistivities to values of 100-300 Ω.m occurred at depths of 2.5-3.5 m with a slight increase at the base of the profile. This drop corresponded to the depth of the base of permafrost in the borehole.

Profile 7 was undertaken in a raised bog on the south side of Cartwright, 100 m from an ocean inlet (Fig. 6d). The profile started in medium-to-high shrubs and scattered White Spruce trees and then traversed an elevated zone of dry peat for

80 m which included four 0.6-1.0 m high rounded mounds separated by shallow depressions. The surface cover was similar to that in profiles 5 and 9, comprised primarily of lichens and exposed peat in the drier areas and low shrubs in the wetter areas. Frost tables were encountered continuously from -22 m to the end of the profile and these corresponded to the drier peat. The median thaw depth was 40 cm (July 25, 2014). Positive resistivity anomalies (> 1000 Ω.m) were present from the surface to a maximum depth of 5.5 m directly beneath the mounds (Fig. 6b) while a layer of low resistivities (< 100 Ω.m)

was present at greater depths. An increase in the modelled values occurred at the base of the profile but in an area of low sensitivity in the model.

The evidence from the field observations is that the mounds in the raised bogs at Cartwright are patches of peatland permafrost and best classified as palsas. The resistivity boundary between frozen and unfrozen soils was inferred to be between 300-400 Ω.m. Based on this value, the greatest thickness of permafrost (5.4 m) was beneath the tallest mound on

profile 5 (P14: 1.3 m high) (Fig. 6a, Table 3 and Table S2), with the maximum value almost the same beneath P21 on profile 7 (5.2 m).  It is not clear whether there were four separate mounds along Profile 7 or local high points on one larger feature, because thin permafrost or deep late-lying seasonal frost was present continuously in the elevated peat (Fig. 6b).

Frozen ground was detected only beneath the palsa surfaces and adjacent elevated (dry) peaty terrain, although late-lying seasonal frost was widespread on some profiles. Given the timing of the fieldwork in late July, it is possible that some

of this persisted through the thaw season. Annual ground temperatures at or close to TTOP at a coastal tundra site and two wetland sites without surface peat near Cartwright ranged from 0.7°C (125 cm depth) to 2.3°C (70 cm depth), showing that permafrost was absent.





### 3.1.4 Main Tickle, NL

The Main Tickle site is a dissected peat plateau complex (0.5-1.0 m high), located 8 km northwest of Cartwright on the northern shore of Sandwich Bay, an inlet of the Atlantic Ocean (Fig. 2c; Fig. 8; Fig. 9). The plateau (Fig. 8c; Fig. 9b) consists of an irregular hummocky surface covering approximately 4 ha within a raised bog that also contains smaller peat

mounds 0.5-1 m high. Surface cover comprised lichens, grasses, other herbs and exposed peat on elevated surfaces while depressions were dominated by sedges and mosses with standing water in some spots. The mean air temperature at the monitoring site on a high point was -1.1°C (2016-2017) while near-surface ground temperatures averaged +0.2°C, both about 1°C lower than at Cartwright. The late-winter snow depth was only 5 cm. The stratigraphy at the borehole, observed during water-jet drilling, was 1.4-1.5 m of peat overlying grey fine sand to coarse silt. Ground temperatures averaged -1.3°C

at 1 m, and -0.7°C at 1.73 m depth, while the permafrost table was around 0.7 m (Fig. 8b). These temperatures are lower than at the other borehole locations but should be regarded with caution as they were measured over only one year.

ERT Profile 12 ran from a shrub-dominated area at the edge of the raised bog across the dissected peat plateau, and terminated in a wetland with shallow surface water (Fig. 8c). A detectable frost table (median depth 58 cm on September 9, 2015) was present across the profile, except at the start and in some isolated depressions (Fig. 8a). Thaw depths increased

towards the end of the profile in an area of shallow surface water, but frozen ground was still present. The ERT profile was interpreted as a two- or three-layer system. Resistivities were lowest at the surface of the profile but still exceeded 500 Ω.m throughout and exceeded 1000 Ω.m in some model blocks. Beneath the surface layer from 32 m to the end of the profile, a layer with resistivities between 800 and 3000 Ω.m was present that is interpreted as permafrost. The basal layer, varying in depth from 5-20 m and with resistivities exceeding 3000 Ω.m, was interpreted as bedrock (which was exposed near the start

of the profile) in an unknown thermal state. Based on these interpretations, permafrost was about 8 m thick at the borehole site, extending well below the base of the borehole whose depth was limited by water supply at the time of drilling.

Profile 13 ran perpendicular to profile 12, traversing most of the peat plateau (Fig. 8c; Fig. 9b). A one m electrode spacing was used to provide greater detail in the near-surface. The median frost table depth on elevated parts of the profile was 53 cm on September 9, 2015 (Fig. 9a). A continuous low resistivity surface layer was present with values below 300

Ω.m (and as low as 20 Ω.m) in depressions where there was no detectable frost table. Five discrete positive resistivity anomalies were present beneath this layer in parts of the profile where the surface was elevated, reaching values of 5000 Ω.m or more. Values beneath depressions were lower, but still exceeded 1000 Ω.m. A high resistivity layer was present across the profile at a depth of approximately 10 m, but most of this constituted an area of low sensitivity in the inversion model.

Permafrost was interpreted to be widespread beneath the peat plateau at Main Tickle. The resistivity boundary for delineating between frozen and unfrozen ground was 400-800 Ω.m but interpretation was complicated by the likely presence of high resistivity bedrock in an unknown thermal state. Permafrost may have been absent between 0-28 m of profile 12 given the lack of a frost table and the probable presence of near-surface bedrock (Fig. 8a). However, permafrost was inferred



to be present throughout the remainder of this profile and all of profile 13 reaching bedrock at a depths of 8-10 m. Resistivities beneath depressions and shallow surface water exceeded 1000 Ω.m suggesting cryotic conditions, but with greater unfrozen moisture contents due to higher temperatures than beneath more elevated parts of the plateau.

### 3.1.5 Neveisik Island, NL

The most northerly wetland on the transect is on Neveisik Island, located about 60 km from the open Atlantic coast at the northeast end of Lake Melville, a tidal estuary fed by the Churchill and other rivers. Peat mounds were common throughout the wetland, but were lower, smaller in areal extent, and more elongated than those observed at the other field sites (Fig. 2d). Vegetation cover was more extensive and higher than at the other sites with many mounds partially covered by low to medium-height shrubs. The mounds showed evidence of degradation, with surface cracks and exposed peat, and numerous ponds were present in the uneven terrain (Fig. 2d).

Ground temperatures on a 1 m high mound about 500 m from the coast averaged -0.9°C at 15 cm depth and -2.3°C at 55 cm (2014-2016). Late-winter snow cover at the site was 15 cm deep. Frost probing at a variety of locations showed thaw depths between 40 and 60 cm for peat mounds (July 31, 2014) with complete probe penetration at non-elevated peat covered or vegetated surfaces. Ground temperatures measured at a low shrub site and a wetland site without dry peat cover on the island, respectively averaged 2.9°C (70 cm depth) and 1.5°C (100 cm depth).

The observations at this wetland indicate that permafrost was present beneath the peat mounds, which are best described as palsas, but was absent elsewhere on the island. The ground temperatures measured beneath the mound are surprisingly low for the isolated patches permafrost zone. They reflect cool summer temperatures in this coastal environment and cold winters during which temperatures as low as -12°C were recorded at a depth of 55 cm.

### 3.2 Peatland permafrost ground ice content

Mound height and permafrost thickness can be used to calculate average excess ice fractions (EICs) in the peatland permafrost using Eq. (1) (Lewkowicz et al. 2011):

(1)

$$Excess\ ice\ fraction = \frac{Mound\ height\ (m)}{Permafrost\ thickness\ (m)}$$

EICs calculated for the palsas examined using ERT ranged from 0.1 to 0.6 with a median value of 0.18 (n=26) (Fig. 10 and Table S2). P6 at Red Bay and the peat plateau at Main Tickle were not included in this analysis because of the difficulty in establishing permafrost thickness where bedrock is inferred to be present at depth. Given the relatively low height of the peat plateau and thick permafrost, the EICs at Main Tickle would likely be lower than 0.15. Mean EICs are lower than those observed for palsas in the southern Yukon (0.2-0.4) (Lewkowicz et al., 2011) but higher than those for



northern Sweden (<0.03-0.25) (Sjöberg et al., 2015) (Fig. 10). The calculated EICs indicate that a considerable amount of heat would need to be transmitted into the mounds to melt the ground ice during permafrost thaw.

### 3.3 Ground temperature modelling

Modelled and measured daily ground temperatures for 2014-2016 were in good agreement at the Blanc Sablon and Cartwright borehole sites following calibration (Fig. S1; Fig. S2). The model reproduced the thin permafrost at both locations and the sizable observed thermal offsets (-2.7°C at Blanc Sablon and -1.7°C at Cartwright). To achieve these matches, the geothermal heat fluxes were set to 0.54 and 1.02 W m$^{-2}$, respectively. These fluxes are an order of magnitude larger than the geothermal heat flux in deep ground (Pollack et al., 1993) but are similar to values used for modelling permafrost mounds in northern Québec (Buteau et al., 2004). They likely reflect advective and vertical heat flow from surrounding unfrozen terrain and water in deep ground rather than being the geothermal heat flow at depth. A higher flux was needed for Cartwright because the observed permafrost thickness is like Blanc Sablon but air and near-surface ground temperatures are lower. This is linked to the smaller and more isolated permafrost mound monitored at Cartwright compared to that at Blanc Sablon. The snow scouring factor was similarly high for the two sites (0.85 for Blanc Sablon and 0.83 for Cartwright) which resulted in modelled winter snow depths of about 15 cm, values that are comparable to the field observations.

Modelled permafrost thickness declined from about 5 m at both sites in 1900 to 3.8 m at Blanc Sablon and 2.5 m at Cartwright in 2016 (Fig. 11). Permafrost thinning due to warming from below began in the 1940s at Cartwright and in the 1950s at Blanc Sablon, but slowed in the 1960s and there was some aggradation in the 1990s. Degradation has occurred since then and is projected to continue. The modelling shows permafrost disappearing at Blanc Sablon during the 2040s or early 2050s with the timing little affected by the choice of RCP (<10 years difference). There is more variation at Cartwright, with degradation in the 2060s under RCP 8.5, in the 2090s under RCP 4.5, and beyond 21$^{st}$ century under RCP 2.6 (Fig. 11b). All these changes in permafrost thickness are due to thaw at the base. A supra-permafrost talik does not develop because the summer thaw depth is always less than the winter freezing depth due to the thin snow cover.

To test whether the permafrost can persist indefinitely if the current climate does not change, we ran the model for Blanc Sablon site with the future climate represented by repeating the climate conditions from 1972 to 2015. We selected this period because it represents the conditions over the past four decades with both cooler and warmer episodes and exactly matches leap years. The results show no future progressive permafrost degradation. This is the case even if the model is initialized without permafrost by artificially increasing daily air temperature in the initialization year (1900) by 2°C. Permafrost establishes in a few years and gradually stabilizes. These results imply that permafrost thermal conditions are in equilibrium with current atmospheric climate and that permafrost would persist at Blanc Sablon if the regional and local climatic conditions of the past four decades were to repeat.

The relative importance of snow wind-scouring and peat cover was assessed by changing these parameters separately for Blanc Sablon. A reduction of 15% in the snow wind scouring factor is sufficient to eliminate contemporary



permafrost at the site. Without any wind scouring, soil temperatures increase by 4.9°C at 0.5-1 m depths and 5.8°C at 5 m depth compared to the calibrated baseline conditions during 1990-2009. Without peat cover, soil temperatures at the same depths increase by 2.4°C and 2.0°C, respectively. This demonstrates that the impact of snow removal by wind on ground temperature is 2-3 times that of the peat cover. In both cases, the surface offset and the thermal offset all become smaller after permafrost disappears. However, the impact of snow wind-scouring on ground temperatures is still two to three times (1.8 and 2.7 times for the depths of 0.5-1 m and 5 m, respectively) that of peat cover.

# 4 Discussion

## 4.1 Southeastern Labrador peatland permafrost

The palsas examined have heights of 0.3-1.5 m which places them at the low end of the circumarctic range (Van Everdingen, 2005), reflecting thin permafrost and excess ice contents generally less than 25%. Peat thickness measured during water-jet drilling varied from 1 to 2 m so the mounds are best-described as mineral-cored palsas (Gurney, 2001). Permafrost thicknesses beneath individual mounds estimated using the ERT results ranged from <1-6 m, although thicknesses of 8-10 m were inferred beneath the peat plateau at Main Tickle. Vegetation on mound crests was limited to ground cover (lichens, grasses and herbs) at almost all sites, with low shrubs present on the sides of some mounds. Bare and wind abraded peat was evident at some sites but the Neveisik Island palsas were the only ones with shrub cover on the mounds themselves.

Measured late-winter snow depths on monitored palsa crests were all less than 25 cm (median of 15 cm) due to wind scouring in winter (Gary Bird, *personal communication*), while snow depths measured at other sites in the region typically exceeded 100 cm. These findings are similar to those from northern Scandinavia where a snow depth of less than about 50 cm was required for contemporary palsa formation (Seppälä, 2011). Given high regional snowfall totals in southeastern Labrador, significant shrub cover on the mound crests would capture more snow (e.g. Way and Lewkowicz, accepted) and the thermal modelling shows that this would lead to the thaw of these permafrost landforms.

Ground temperatures recorded at TTOP on the peatland permafrost features varied from -0.7°C at the southernmost site (Blanc Sablon) to -2.3°C at the northernmost one (Neveisik Island) showing a clear latitudinal gradient (Fig. 12). These are relatively low values given the range of MAAT, and are due to the small surface offsets and large thermal offsets measured at the study sites. The latter could be caused by the slow transient response of ground temperatures to recent regional warming (Jorgenson et al., 2010; Lewkowicz et al., 2016; Morse et al., 2016), but we favour the explanation that these large offsets are due to differences in frozen and unfrozen thermal conductivities resulting from thick peat found in the well-drained raised bogs (Seppälä, 2011). This inference is supported by very shallow thaw depths, ranging from 37-65 cm (median: 46 cm) and by the modelling that indicates that the current ground temperatures are in equilibrium with the climate.

The shallow thaw depths observed also run counter to broader scale models (e.g. Smith and Riseborough, 2002; Zhang et al., 2008b) which predict thick active layers (1-2 m) at the southern end of the discontinuous zone. However, coarse

earth system models do not consider that permafrost in this zone, where present, is typically in peatlands which are not represented well on most national scale surficial maps used as modelling inputs.

An unusual feature of the thermal conditions at the borehole sites is the relatively thin permafrost associated with strong thermal gradients. Mean temperatures at 1 m depth at the four sites with boreholes varied from -0.7°C to -1.4°C, while permafrost thickness based on drill records and the ERT profiles varied from 3.7 to 8 m. Calculated geothermal gradients ranged from 0.10°C/m to 0.38°C/m which greatly exceed typical regional values of 0.005-0.01°C/m (Grasby et al., 2012). These results imply high heat flows into the atmosphere and suggest that these small permafrost bodies act as conduits for advective heat originating from surrounding snow-covered terrain in winter and from the unfrozen bog in summer. Thermal modelling for two of the borehole sites could not match the measured ground temperatures without prescribing an order of magnitude increase in the geothermal heat fluxes compared to typical values for extensive permafrost. Thus, local conditions are critical for determining permafrost conditions, especially in the isolated patches zone. This underlines the challenges to map permafrost at high spatial resolution and to predict the changes of permafrost to climate change.

It has been hypothesized that palsas near Blanc Sablon may be relict features that have persisted in the area since the Little Ice Age due to coastal effects, including persistent cloudiness (Dionne, 1984; Dionne and Richard, 2006). Elsewhere, there is evidence that permafrost in the coastal zone has degraded over the past half-century. For example, palsas at Cartwright were 1.5 m high and peat plateaus were > 60 m long and > 6 m wide in 1968 (Brown, 1979, 1975). Currently, they do not exceed 1 m in height and permafrost-cored terrain appears to cover a much smaller part of the landscape than 50 years ago. Similarly, palsas and peat plateaus at Neveisik Island were 1.8 m high and covered most of the bog in 1968 (Brown, 1979, 1975), whereas we observed no mounds greater than 1 m high or with long axes >20 m, and many collapse scars and thermokarst ponds (Fig. 2c). Degradational features on palsas were also evident at Blanc Sablon and Cartwright where several mounds exhibited large cracks exposing their interiors. Ground temperatures recorded by Allard et al. (2014) between 1991-2012 (with a data gap from 1998-2004) at a palsa field north of Blanc Sablon show that temperatures at 50 cm depth increased to the point where average ground temperatures regularly exceed 0°C (Fig. S3). These data support our numerical modelling which indicates past thinning of permafrost and the general field interpretation that regional permafrost degradation has occurred (Way and Lewkowicz, 2016).

### 4.2 Permafrost in southeastern Labrador

A preliminary analysis of Google Earth™ imagery and observations made by Brown (1979, 1975) suggest that peatland permafrost features may exist elsewhere in the coastal barrens between Blanc Sablon and Cartwright. A detailed inventory of current palsas and peat plateaux, however, would require high resolution aerial photography or sub-meter resolution satellite imagery to identify the very small mounds found in many areas (Borge et al., 2017) and this has not yet been undertaken. Satellite imagery does show widespread thermokarst ponds in the region, including on the northernmost tip of Newfoundland. Archeological investigations at L'Anse aux Meadows (51.6°N) described the excavation of a palsa with

thick surface peat (Henningsmoen, 1977), and given the similar geographic setting and climate in northernmost parts of Newfoundland to sites at the southern end of our transect, there is no reason *a priori* that palsas could not exist there.

Peatland permafrost showed much smaller surface offsets (median: 1.1°C ) and much larger thermal offsets (median: -2.3°C) than other land cover types where permafrost is absent (Way and Lewkowicz, accepted). These results indicate that peatland permafrost is ecosystem protected (Shur and Jorgenson, 2007) and only tenable due to the elevated surface, low vegetation cover and high wind speeds which facilitate low snow depths, and the thick peat which produces a large thermal offset (e.g. Jorgenson et al., 2010; Lewkowicz et al., 2016). The modelling for Blanc Salon indicates that this ecosystem protection will become insufficient during climate warming, as the apparent thermal offset grows to about -3.5°C just prior to permafrost disappearance in response to an increasing disequilibrium between the still-frozen ground and the mean surface temperature (Fig. S4).

The peatland permafrost features were all located in the coastal barrens with MAATs between -0.5°C and +1.0°C (2013-2016). These coastal climates are influenced by seasonal sea ice promoting near-surface temperature inversions that cool the ground in winter and spring (Atkinson and Gajewski, 2002; Maxwell, 1981; Smith and Bonnaventure, 2017). However, additional factors must play a role because permafrost has not been observed in colder areas inland or at higher elevation (Brown, 1979; Way and Lewkowicz, accepted) (Fig. 12). First, when all other factors are the same, mean ground temperatures should be lower in coastal areas with smaller temperature ranges (Riseborough, 2004). Second, melting sea ice and ocean-atmosphere contrasts promote coastal fog and persistent cloudiness in summer, leading to Arctic-like temperatures and lower amounts of insolation along the outer coasts in Labrador (Maxwell, 1981; Way et al., 2014, 2017). Finally, persistent high winds and frequent winter melt events likely modify snow properties (e.g. density) which increase winter heat exchange between the ground and the atmosphere (Derksen et al., 2014). In addition, windy conditions influence snow redistribution and enhance wind scouring which our modelling shows is critical for permafrost. However, the absence of permafrost beneath wind-scoured tundra hilltops near Blanc Sablon, Cartwright Junction and Cartwright demonstrates that permafrost cannot now exist in mineral soils at low elevations in southeastern Labrador. Towards the northern end of the study transect, larger and thicker permafrost bodies were present near Cartwright and Main Tickle (Fig. 12) and wetland terrain adjacent to the peat plateau remained frozen near the end of the thaw season. These results suggest that the -1°C MAAT threshold commonly used in northern Scandinavia (e.g. Vorren, 2017) for delimiting the upper climatic boundary for palsas is too low for Labrador, and that our results resemble those from a recent survey in Finnmark which found permafrost patches at MAATs as high as +1°C (Borge et al., 2017).

**4.3 Permafrost persistence**

Our thermal modelling indicates that peatland permafrost could thaw in most of southeastern Labrador in the 21st century, with southern sites losing permafrost by about 2050. Farther north, responses depend on the RCP scenario and there is the possibility of permafrost persistence if RCP 2.6 is followed. However, the modelling considered only vertical fluxes whereas isolated patches of peatland permafrost may undergo degradation via two-dimensional or three-dimensional



processes including enhanced advective heat transfer, wind abrasion of surface peat, and block collapse into adjacent standing water, all of which may accelerate the rate of permafrost loss. Therefore, the modelling results may be optimistic in terms of permafrost persistence.

The modelling shows that wind scouring of snow is more important than peat cover and that, unlike in other studies (Camill and Clark, 1998; Halsey et al., 1995; Morse et al., 2016; Zhang et al., 2008a), the thermal condition of the permafrost is in equilibrium with the current atmospheric climate. However, the wind scouring occurs because of the raised elevation of the palsa crest, which is itself the product of ice segregation during permafrost formation, and because of the absence of tall shrubs. Modelling showed that even a slightly thicker snow cover (still below half the average snow depth) would preclude the presence of permafrost and hence would make it impossible for a mound to start unless its geographic or topographic position favoured extreme wind scouring. Therefore, in most of the locations investigated in this study the peatland permafrost is relict because although its thermal condition is in equilibrium with the current climate, the landforms could not develop under current conditions.

Permafrost degradation is often associated with the development of near-surface taliks in the field (Romanovsky et al., 2010; Shaoling et al., 2000) and in model simulations (Delisle, 2007; Schaefer et al., 2011; Shaoling et al., 2000; Zhang, 2013; Zhang et al., 2008b). Our modelling did not show talik development during degradation because winter freezing is much greater than summer thaw due to the combination of snow scouring by wind and thick peat cover. At our sites, the permafrost only thinned upwards from the base which is not the norm for deep permafrost. The modelling also showed that permafrost persistence and degradation is more complicated than predicted by coarse grid models. Even a relatively fine grid model (e.g. Way and Lewkowicz, 2016) has difficulties including local conditions which may be crucial for permafrost persistence. These results emphasize the importance of considering local snow conditions for permafrost modelling and mapping and suggest that the southern boundary of permafrost may not move as quickly as regional models predict (e.g. Koven et al., 2013). The impact of differing climate scenarios also depends on permafrost conditions. At the warmer Blanc Sablon site, permafrost degradation is predicted to be similar for all three RCP simulations, while at the colder Cartwright site, there are significant differences in the timing of degradation and even the prediction of permafrost persistence under RCP2.6.

## 5 Conclusion

Isolated patches of peatland permafrost in the coastal zone of southeast Labrador extend as far south as 51.5°N on the Gulf of St. Lawrence. These patches take the form of mineral-cored palsas and peat plateaux and are mainly present in raised bogs where air temperatures range from +1°C in the south to -1.1°C at 53.7°N. Field observations show that both thin wind-scoured snow cover on mound crests causing small surface offsets and thick peat causing large thermal offsets are crucial to permafrost existence. Consequently, we found no evidence that permafrost is present at low elevations outside peatlands in this part of the isolated patches zone, even where air temperatures are substantially lower.



Permafrost thicknesses within peatlands were all less than 10 m which was lower than expected based on measured ground temperatures at the top of permafrost of -0.7°C to -1.4°C. We attribute the high geothermal gradients to the three-dimensional configuration of the isolated permafrost bodies which would promote advective heat flow from surrounding non-permafrost areas, as well as possible heat transport from sub-permafrost groundwater. One-dimensional thermal modelling for two sites could not accurately simulate the measured ground thermal regimes without prescribing elevated geothermal heat fluxes. Sensitivity modelling showed that the reduced snow cover had an impact of 2-3 times that of the loss of peat cover at these sites.

Modelling results show that the thermal condition of the permafrost is in equilibrium with the current climate but the permafrost is dependent on thin snow, which would be unlikely to exist if a mound were not present to cause wind scouring or in very particular topographic positions. Despite this, our modelling indicates that features at the northern end of the transect could persist until the end of the 21[st] century under RCP 4.5 and 2.6, but would degrade entirely under RCP8.5 by about 2060. Peatland permafrost at the southern end of the transect is more vulnerable and will degrade under all three scenarios by about 2050. Given the sensitivity of southeastern Labrador peatland permafrost to future changes and potential disturbances, more effort is needed to locate, characterize and monitor these ecosystem-protected permafrost occurrences.

*Data availability.* All datasets used in this manuscript are available from the corresponding author on request. Borehole temperature data will be submitted to GTN-P following acceptance of the relevant papers in peer reviewed journals. Code used to compile information from monitoring stations is available in the supplemental materials of Way and Lewkowicz (accepted).

*Competing interests.* The authors declare that they have no conflict of interest.

*Acknowledgements.* Financial support for the research was provided by NSERC, the Northern Scientific Training Program, the University of Ottawa and the Royal Canadian Geographical Society. Additional support to RGW was provided by the W. Garfield Weston Foundation. We are grateful to Alexander Brooker, Maxime Duguay and Caitlin Lapalme for their assistance in the field and to Patricia Way, Gary Bird, George Way and Brenda Way for their logistical support. We thank Cartwright community members including Gary Bird, Leslie Hamel and Lloyd Pardy for their help in identifying suitable study sites. The manuscript was improved through discussions with Michel Allard, Denis Lacelle, Sharon Smith, Bruce Roberts and Darya Anderson. YZ's work was funded by Polar Knowledge Canada Science and Technology Program (project 186). It also contributes to a project affiliated with the Arctic Boreal Vulnerability Experiment (ABoVE), a NASA Terrestrial Ecology Program.




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

40

45



**Tables**

**Table 1.** Site information and field data collected (2013-2017).

| Site location(s) | Latitude | Longitude | Elevation | Site description | Data collected | ERT ID |
|---|---|---|---|---|---|---|
| Blanc Sablon, QC | 51.4565°N | 57.1185°W | 115 m a.s.l. | Palsa field | Air and ground temperatures to 4.2 m; two ERT surveys; one ibutton stake (10 loggers) | 1 & 2 |
| Blanc Sablon, QC | 51.4577°N | 57.1219°W | 143 m a.s.l. | Barren hilltop | Ground temperatures to 0.62 m (two locations) | |
| Red Bay, NL | 51.7589°N | 56.4135°W | 72 m a.s.l. | Palsa field | Air and ground temperatures to 4.3 m; ERT survey; one ibutton stake (10 loggers) | 3 |
| Red Bay, NL | 51.7581°N | 56.4143°W | 77 m a.s.l. | Peat-covered bedrock | ERT survey | 4 |
| St. Charles River, NL | 52.1611°N | 56.0977°W | 217 m a.s.l. | Dense forest | Air and ground temperatures to 0.7 m | |
| St. Charles River, NL | 52.1385°N | 56.1347°W | 333 m a.s.l. | Low-shrub tundra | Air and ground temperatures to 0.8 m | |
| Cartwright Junction, NL | 53.0640°N | 57.6796°W | 190 m a.s.l. | Open forest | Air and ground temperatures to 1.0 m; one ibutton stake (10 loggers) | |
| Cartwright Junction, NL | 53.0739°N | 57.6649°W | 302 m a.s.l. | Barren hilltop | Air and ground temperatures to 0.6 m; one ibutton stake (10 loggers) | |
| Cartwright, NL | 53.7037°N | 57.0098°W | 5-10 m a.s.l . | Palsa fields | Air and ground temperatures to 2.2 m and 5.7 m; seven ERT surveys; one ibutton stake (10 loggers) | 5 to 11 |
| Cartwright, NL | 53.7255°N | 56.9641°W | 158 m a.s.l. | Barren hillslope | Air and ground temperatures to 0.9 m; one ibutton stake (10 loggers) | |
| Main Tickle, NL | 53.7357°N | 57.1331°W | 9 m a.s.l. | Peat plateau | Air and ground temperatures to 1.7 m; two ERT surveys; one ibutton stake (10 loggers) | 12 & 13 |
| Neveisik Island, NL | 53.9851°N | 58.8358°W | 12 m a.s.l. | Palsa field | Air and ground temperatures to 0.6 m; one ibutton stake (10 loggers) | |
| Neveisik Island, NL | 53.9851°N | 58.8358°W | 3-12 m a.s.l. | Open forest | Ground temperatures to 1.0 m (two locations) | |



**Table 2.** Site information for temperature monitoring boreholes.

| Borehole ID | Elevation (m) | Location | Date established | Measurement depths (m) | Logger type |
|---|---|---|---|---|---|
| WJD01 | 11 | Cartwright, NL | 2014.08.09 | 0.25; 0.5; 1.5; 2.15 | Ibutton |
| WJD02 | 14 | Cartwright, NL | 2014.07.23 | 0.5; 1.0; 2.0; 3.0; 4.25; 5.7 | Hobo Pro V2 |
| WJD03 | 115 | Blanc Sablon, QC | 2014.08.05 | 0.25; 0.5; 1.0; 2.0; 3.0; 4.2 | Hobo Pro V2 |
| WJD04 | 75 | Red Bay, NL | 2014.08.06 | 0.25; 0.5; 1.0; 2.0; 3.0; 4.25 | Hobo Pro V2 |
| WJD05 | 8 | Main Tickle, NL | 2015.09.09 | 0.75; 1.0; 1.25; 1.73 | Hobo Pro V2 |



**Table 3.** ERT surveys and inferred permafrost thickness.

| ERT profile | Length / Minimum electrode spacing | Vegetation description (highest to lowest coverage) | Inferred maximum permafrost thickness |
|---|---|---|---|
| 1 | 100 m / 1 m | Bare peat, lichens, grasses, low shrubs; | 6 m |
| 2 | 50 m / 0.5 m | Bare peat, lichens, grasses | 6 m |
| 3 | 80 m / 1 m | Sedges & grasses, medium height shrubs, lichens, bare peat | 3.5 m |
| 4 | 80 m / 1 m | Bare peat, sedges and grasses, lichens | 6 m; bedrock possibly frozen deeper |
| 5 | 240 m / 2 m | Medium height shrubs, bare peat, sedges & grasses, lichens | 5.4 m |
| 6 | 80 m / 2 m | Lichens, medium height shrubs, sedges & grasses, bare peat | 3.9 m |
| 7 | 120 m / 2 m | Lichens, bare peat, low shrubs, grasses | 5.2 m |
| 8 | 40 m / 1 m | Lichens, grasses, bare peat | 4.8 m |
| 9 | 40 m / 1 m | Lichens, bare peat, grasses, low shrubs | 3.7 m |
| 10 | 80 m / 2 m | Sedges & grasses, lichens, bare peat | 3.7 m |
| 11 | 40 m / 1 m | Sedges & grasses, lichens, bare peat | 0.9 m |
| 12 | 160 m / 2 m | Lichens, bare peat, grasses, medium height shrubs | 8 m; bedrock possibly frozen deeper |
| 13 | 100 m / 1 m | Lichens, bare peat, sedges & grasses | 8 m; bedrock possibly frozen deeper |

10 **Figures**



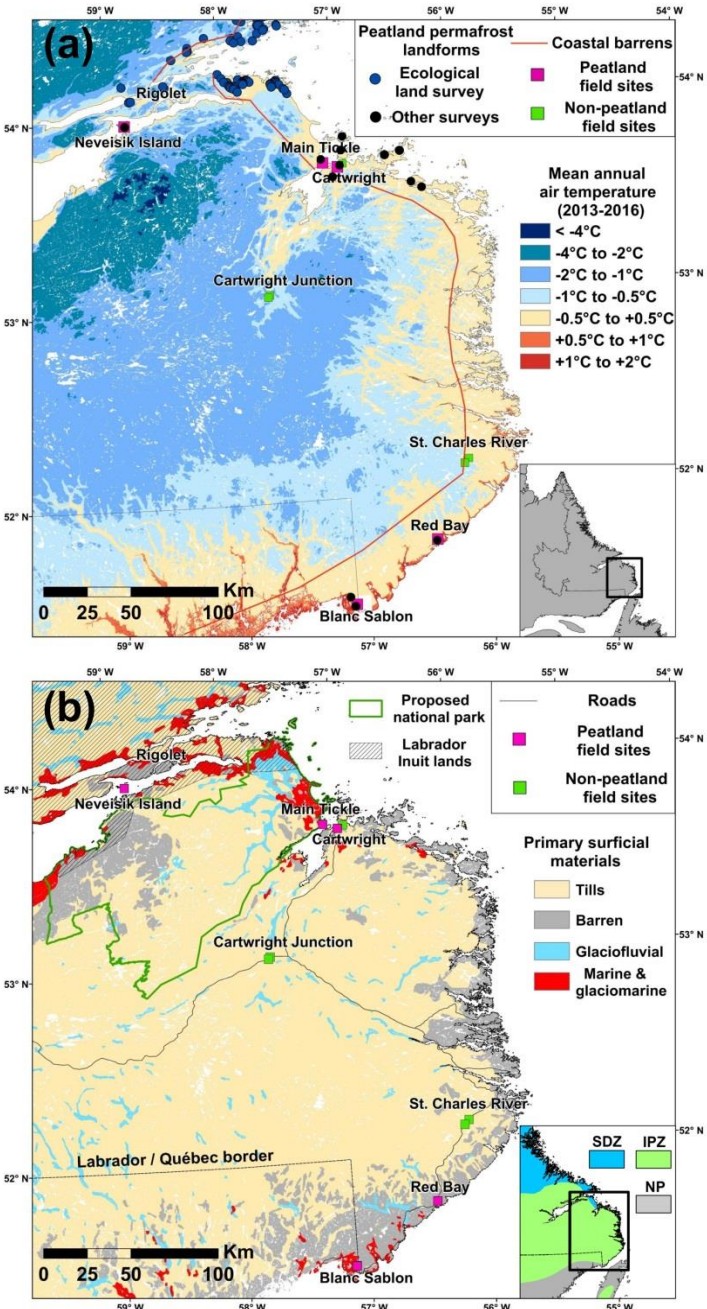

**Figure 1.** Location of peatland and non-peatland study sites in southeastern Labrador in relation to: (a) gridded (100 m resolution) mean annual air temperatures (2013-2016) (Way et al., 2017) and peatland landforms mapped in various studies, and (b) surficial materials (Fulton, 1995). The inset map in (a) shows the study location in eastern Canada and the boundary of the coastal barrens ecozone (Roberts et al., 2006) is shown as an orange line. Labrador Inuit lands and the proposed Mealy Mountains National Park are outlined in (b). The inset in (b) shows the boundaries of permafrost zones (after Heginbottom et al., 1995) where SDZ = sporadic discontinuous zone; IPZ = isolated patches zone; and NP = no permafrost.





**Figure 2.** Low altitude UAV imagery of peatland permafrost sites at (a) Blanc Sablon, QC; (b) Red Bay, NL; (c) Main
Tickle, NL; and (d) Neveisik Island, NL. White arrows point to examples of permafrost-cored mounds.







**Figure 3.** Peatland permafrost at Blanc Sablon. Frost tables and modelled resistivities on (a) ERT profile 1; and (b) ERT profile 2. Depths with permafrost in the borehole are denoted by hatching; (c) Air and ground temperature envelopes (2015-2017); (d) UAV vertical image of the site. Black lines delimit the approximate location of the ERT survey lines and text shows the start and end points.



**Figure 4.** Peatland permafrost at Red Bay. (a) Frost tables and modelled resistivities on ERT profile 3. Depths with permafrost in borehole WJD04 are denoted by hatching; (b) Air and ground temperature envelopes (2015-2017); (c) UAV vertical image of the site. Black line delimits the approximate location of the ERT survey line and text shows the start and end points.





**Figure 5.** Peatland permafrost at Red Bay. (a) Frost tables and modelled resistivities on ERT profile 4; (b) Oblique aerial photo upslope along ERT profile 4 from the survey start point. Dotted black line delimits location of ERT survey line; (c) UAV vertical image of the site. Black line delimits the approximate location of the ERT survey line and text shows the start and end points.



**Figure 6.** Peatland permafrost at Cartwright. Frost tables and modelled resistivities on (a) ERT profile 5; and (b) ERT profile 7. UAV vertical images of (c) ERT profile 5 and (d) ERT profile 7, with black lines delimiting the approximate location of the ERT survey lines and text showing the start and end points.





**Figure 7.** Peatland permafrost at Cartwright. (a) Frost tables and modelled resistivities on ERT profile 9. Depths with permafrost in borehole WJD02 are denoted by hatching; (b) Air temperature (from nearby station AMET13) and ground temperature envelopes (2015-2017); (c) UAV vertical image of the site. Black line delimits the approximate location of the ERT survey line and text shows the start and end points.



**Figure 8.** Peatland permafrost at Main Tickle. (a) Frost tables and modelled resistivities on ERT profile 12. Depths with permafrost in borehole WJD05 are denoted by hatching. Note: the colour scale differs from that in Figures 3-7 to differentiate between zones of high resistivity. Consequently, some green zones on this modified scale are inferred to be frozen; (b) Air and ground temperature envelopes (2016-2017); (c) UAV vertical image of the site. Black line delimits the approximate location of the ERT survey line and text shows the start and end points.



**Figure 9.** Peatland permafrost at Main Tickle. (a) Frost tables and modelled resistivities on ERT profile 13. Note: the colour scale differs from that in Figures 3-7 to differentiate between zones of high resistivity. Consequently, some green zones on this modified scale are inferred to be frozen; (b) UAV vertical image of the site. Black line delimits the approximate location of the ERT survey line and text shows the start and end points.



**Figure 10.** Comparison between maximum mound height and inferred maximum permafrost thickness. Dotted lines depict the association for excess ice quantities of 10%, 20%, 30% and 40%.



**Figure 11.** Permafrost thickness modelled with NEST for 1900-2100 for (a) Blanc Sablon; and (b) Cartwright. Future predictions are for RCP2.6, RCP4.5 and RCP8.5.



**Figure 12.** Average ground temperatures (2013-2016) at or close to the top of permafrost or the base of seasonal freezing at University of Ottawa monitoring locations in southeastern Labrador superimposed on a map of mean annual air temperature (2013-2016) **(Way et al., 2017)**. Letters denote primary land cover class at each location: F – Forest; P – Palsa; PP – Peat Plateau; S – Shrub; T – Tundra; and W – Wetland. At Cartwright, ground temperatures for palsas were averaged from three sites. Mealy Mountains data (indicated by [1]) were collected by Jacobs et al. (2014) between 2001-2009.