# Peer review of "Characteristics and fate of isolated permafrost patches in coastal Labrador, Canada"

_The Cryosphere, 2017_

## Referee Comment (RC1) · Anonymous Referee #1 · 5 Feb 2018

General comments: The manuscript describes characteristics of palsa peatland permafrost in coastal Labrador, Canada, derived from electrical resistivity studies and local observations of ground temperature and climatic variables. These descriptions, from several sites in the area, are complemented with numerical transient modeling of the fate of permafrost under future warming for two studied boreholes in the study area. The results show that thin permafrost exist in isolated patches in palsa peatlands in the relatively warm coastal area of Labrador (-1.1° to 1° C average annual air temperature) and that it is in equilibrium with present climate due to a combination of thin snow cover and large thermal offset caused by the peat cover. The simulations show that the permafrost would degrade in most or all of the study area for the range of tested climate warming scenarios.

[Figure]

The dynamics of permafrost in the discontinuous, sporadic and isolated patches permafrost zones is challenging to predict with today's simulation tools, which generally focus on coarse scales and vertical heat fluxes. This study adds valuable information on how and where permafrost appears in isolated patches, which is needed for understanding of how climatic changes can affect these areas. It further addresses the challenges associated with using numerical permafrost models, which do not represent lateral heat fluxes, for simulating this type of permafrost.

The manuscript is well-written and structured. The introduction is brief but relevant. The methods section could need some expansion and clarification on some details, in particular the simulation procedure should be better described (see detailed comments below). The results are presented straight-forwardly. The discussion puts the results in context of current knowledge and highlights the relevance and impact of the findings, but I lack a mention of the implications of the assumptions that the modeling is based on (see detailed comments below). The manuscript contains many figures which I think is of value for a study that presents this type of geophysical data for describing permafrost.

As the manuscript presents a significant contribution to our current understanding of palsa and peat plateau permafrost characteristics and dynamics, and is generally well-written, I recommend that it is accepted after minor revisions.

Detailed comments:

P4, L1: Way and Lewkowicz full citation should be available when this is published.

P4, L7: What is the water-jet method?

P4, L25: What is meant by "low sensitivity areas (<0.1)" and how does it reflect uncertainties?

S1: In general, it would be good to state what the different parameter values are based on (local data, literature...?)

"Degree of decomposition" increases from 0.1 to 0.4 – is this linearly with depth?

"Organic matter content" – same as above

"Degree of decomposition" again. . . Is this for mineral substrate?

"Fraction of quartz" and "Thermal conductivity of rock" – how were these values chosen?

"Geothermal heat flux" – these were calibrated, right? This should be clearly stated in the table.

"Water table reduces 10% when above ground surface" – This sentence is really difficult to understand. I have no idea what it means. 10 % of what? In what way does the model include water above the ground surface, and how is a lateral flux of water incorporated?

The vegetation type is listed as shrub, but in your vegetation description you state that no shrubs are present on the palsas, only on the sides. How is this choice motivated?

P4, L33: Please clarify that all thermal properties and other necessary parameters and their motivations (literature, field observations) are listed in S1 and make sure that they are.

P4, L30 – P5, L12: A presentation of the model discretization/mesh is lacking. It is also unclear how initial conditions were set up and how/if any spin-up procedures were performed. Was the model parameters calibrated after a spin-up from year 1900? If so, what were the initial conditions at year 1900? Was daily air temperature the only data needed for running the model for all time periods? If there is a snow wind-scouring factor, I would assume that the model also takes in snow/precipitation data. Please formulate this more clearly than "climatic inputs" (P4, L8).

P6, L10-12: I do not understand how the accuracy of the loggers and the inherent uncertainty in ERT is considered in estimating this very precise thickness value without

an uncertainty range. Is this an estimate of maximum likely thickness?

P8, L30: TTOP?

P11, L4-15: So, the higher magnitude of the geothermal heat flux is used to compensate for the lack of horizontal heat fluxes in the model. How can you assume that the influence of the horizontal fluxes is stable over time, i.e. for keeping the calibrated geothermal heat flux values steady over the simulated warming periods? I understand that it is probably not possible to test this within this study, but the importance of this assumption should be noted in the text, especially as you argue for a higher geothermal heat flux in Cartwright due to smaller palsas in this location.

P11, L9: Do you mean horizontal (instead of vertical)?

P12, L20: Another relevant reference about how snow influences palsa ground temperatures in Scandinavia is Sannel et al., 2016:

Sannel, A.B.K., Hugelius, G., Jansson, P., Kuhry, P., 2016: Permafrost warming in a subarctic peatland – which meteorological controls are most important? Permafrost and Periglacial Processes, doi:10.1002/ppp.1862.

P12, L26-30: Why could it not be both? The simulation tool applied here does not take into account potential feedback processes that could speed up warming/thawing with time, such as increases in lateral heat transport as permafrost bodies decrease in size, and feedback from changing topography (with melt of ground ice) to decreased wind-scouring and subsequent warming/thawing.

P13, L8: Why does these heat flows need to be advective? See for example Kurylyk et al., 2016:

Kurylyk, B. L., M. Hayashi, W. L. Quinton, J. M. McKenzie, and C. I. Voss (2016), Influence of vertical and lateral heat transfer on permafrost thaw, peatland landscape transition, and groundwater flow, Water Resour. Res., 52, 1286–1305, doi:10.1002/2015WR018057.

Fig. S4: This figure needs some more explanation. It would be helpful if a definition of the thermal offset was given, that was used for these calculated values before and (most of all) after permafrost thaw.

P15,L1-3: A couple of references of relevance in a discussion on the relative importance of these processes when modeling permafrost in palsa peatlands are the above listed article by Kurylyk et al. (2016) and an article by Sjöberg et al. (2016). In both publications, numerical models that couple heat and water fluxes in 2D (thus incorporating advective and lateral diffusive heat fluxes) are applied for similar environments. Kurylyk et al. find that horizontal thaw rates are much higher than vertical thaw rates, for a simulated peat plateau and that most of this is due to horizontal diffusive heat fluxes. Sjöberg et al. studied lateral heat fluxes through a permafrost-free fen separating two peat plateau, and found that incorporating advective heat flow in the model changed the spring thaw date of the fen by a month.

Sjöberg, Y., E. Coon, A. B. K. Sannel, R. Pannetier, D. Harp, A. Frampton, S. L. Painter, and S. W. Lyon. (2016), Thermal effects of groundwater flow through subarctic fens: A case study based on field observations and numerical modeling, Water Resour. Res., 52, 1591–1606, doi:10.1002/2015WR017571.

---

## Referee Comment (RC2) · Anonymous Referee #2 · 19 Feb 2018

The manuscript describes detailed field studies of isolated bodies of permafrost along a transect in southeastern Labrador, Canada, as well as corresponding numerical simulations. I agree with reviewer #1 what the manuscript is generally well written with a clear structure and adds significant new knowledge to our understanding of isolated permafrost bodies in peatlands, but with more information needed especially on the modelling part of the study. I also recommend publication after minor revision, with the following additional comments to be addressed:

P1, L13: consider adding "in this region" after permafrost, as it might otherwise sound like thick peat is generally critical to permafrost.

P1, L17: Here you mention "downscaled global warming scenarios", but there is no mentioning of downscaling in the methods section, only using multi model mean values.

P3, L13-19: how does this relate to the information in P2L27, that regional air temperatures have been rapidly increasing over the last 50 years? Was the study period colder than for instance the mean of the last decade?

P3, L13-19: consider adding some more information here about the climatic conditions, like mean annual precipitation.

P5, L6-12: which parameters were calibrated and how should be more clearly stated.

P5, L8-9: Is this the multi-model mean from the CMIP5 archive? Were these values used directly, or just the trend? If these were used directly, how did the values correspond to the measurements in the overlap period (e.g. 2006-2016)?

P6, L13-15: Here and elsewhere (e.g. P6, L30) the authors describe more (seasonal) ice than expected. Is this an indication that the study period was colder than the previous years (see comment P3, L13-19)?

P7, L20: Drop "s" in "Tables 3".

P11, L9-10: I find the explanation for the high geothermal heat fluxes needed reasonable. However, if this is really heat flow from the surroundings, is it reasonable to keep this constant throughout the simulation? Also, what is the error introduced by adding this heat at the base of a 120m soil column?

P12, L23: TTOP should be explained here. What is it and how is it derived? If these are values derived with the TTOP model I would not call these "recorded".

P15, L2-3: I would add the snow feedback to the reasons why these simulations might be too optimistic: When the PF thaws (and excess ice melts) less snow should be removed, and the wind-scouring factor should decrease, which is not accounted for here.

P15, L22: Koven et al. (2013) does not describe regional model simulations, but global.

Table 3: It would be useful to have the locations in this table as well, so one would not

have to go back and forth between this table and table 1. Consider adding this as an extra column here or naming the ERT profiles according to the locations (e.g. BS1, BS2, RB1, RB2 etc).

---

## Author Response (AR1)

**Characteristics and fate of isolated permafrost patches in coastal Labrador, Canada**

Robert G. Way[1,2], Antoni G. Lewkowicz[2], Yu Zhang[3]

[1]Labrador Institute, Memorial University of Newfoundland, Happy Valley-Goose Bay, A0P1E0, Canada
[2]Department of Geography, Environment and Geomatics, University of Ottawa, Ottawa, K1N6N5, Canada
[3]Canada Centre for Mapping and Earth Observation, Natural Resources Canada, Ottawa, K1A0E4, Canada

*Correspondence to*: Robert G. Way (rway024@uottawa.ca)

**Reviewer responses**

**Reviewer 1**

General comments: The manuscript describes characteristics of palsa peatland permafrost in coastal Labrador, Canada, derived from electrical resistivity studies and local observations of ground temperature and climatic variables. These descriptions, from several sites in the area, are complemented with numerical transient modeling of the fate of permafrost under future warming for two studied boreholes in the study area. The results show that thin permafrost exist in isolated patches in palsa peatlands in the relatively warm coastal area of Labrador (-1.1°C to 1°C average annual air temperature) and that it is in equilibrium with present climate due to a combination of thin snow cover and large thermal offset caused by the peat cover. The simulations show that the permafrost would degrade in most or all of the study area for the range of tested climate warming scenarios.

The dynamics of permafrost in the discontinuous, sporadic and isolated patches permafrost zones is challenging to predict with today's simulation tools, which generally focus on coarse scales and vertical heat fluxes. This study adds valuable information on how and where permafrost appears in isolated patches, which is needed for understanding of how climatic changes can affect these areas. It further addresses the challenges associated with using numerical permafrost models, which do not represent lateral heat fluxes, for simulating this type of permafrost.

The manuscript is well-written and structured. The introduction is brief but relevant. The methods section could need some expansion and clarification on some details, in particular the simulation procedure should be better described (see detailed comments below). The results are presented straight-forwardly. The discussion puts the results in context of current knowledge and highlights the relevance and impact of the findings, but I lack a mention of the implications of the assumptions that the modeling is based on (see detailed comments below). The manuscript contains many figures which I think is of value for a study that presents this type of geophysical data for describing permafrost.

As the manuscript presents a significant contribution to our current understanding of palsa and peat plateau permafrost characteristics and dynamics, and is generally well written, I recommend that it is accepted after minor revisions.

**[Authors' response] The authors' would like to thank reviewer 1 for taking the time to review this manuscript. We agree with reviewer 1's comments and have amended the manuscript accordingly.**

Detailed comments:

P4, L1: Way and Lewkowicz full citation should be available when this is published.

**[Authors' response] We agree. This contribution is now published online and we have amended the citation accordingly.**

P4, L7: What is the water-jet method?

**[Authors' response] Text added:**

**Five shallow boreholes (Table 2) were drilled using water jet drilling with a low horsepower pumping water from a nearby water body down a steel pipe used for penetrating the ground. Immediately post-drilling, holes were cased with 1-inch PVC pipe.**

P4, L25: What is meant by "low sensitivity areas (<0.1)" and how does it reflect uncertainties?

**[Authors' response] Amended to:**

**Modelled resistivities are presented as model blocks with less reliably measured blocks (sensitivity values < 0.1) faded to reflect that these sections are less certain in the modelled profile.**

S1: In general, it would be good to state what the different parameter values are based on (local data, literature. . .?)

**[Authors' response] Good suggestion. We added a column in the table S1 to indicate the sources and methods that each parameter was based on.**

"Degree of decomposition" increases from 0.1 to 0.4 – is this linearly with depth?

**[Authors' response] Yes. The degree of decomposition slightly increases with depth. Since the numbers are defined by the model and it is hard for readers to understand it, we revised it into descriptive words (texture of the peat).**

"Organic matter content" – same as above

**[Authors' response] Revised. For Cartwright site: 1.2-1.5 m: decreases from 100% to 5%, 1.5-3.2 m: 5%, then linearly decreases to 1% at 10 m. For Blanc Sablon: 1.75-2.0m: decrease from 100% to 5%, 2.0-3.2 m: 5%, then linearly decreases to 1% at 10 m.**

"Degree of decomposition" again. . . Is this for mineral substrate?

**[Authors' response] Yes. We revised it as "Texture of the organic matter in mineral soils". The numbers in the table were revised as descriptive words (from hemic to well decomposed at depth).**

"Fraction of quartz" and "Thermal conductivity of rock" – how were these values chosen?

**[Authors' response] The references were added for them.**

"Geothermal heat flux" – these were calibrated, right? This should be clearly stated in the table.

**[Authors' response] Revised as suggested.**

"Water table reduces 10% when above ground surface" – This sentence is really difficult to understand. I have no idea what it means. 10 % of what? In what way does the model include water above the ground surface, and how is a lateral flux of water incorporated?

**[Authors' response] Although the model is one dimensional, we simply parameterized lateral water flows based on modelled water table. In our case, for example, when water table is above the land surface, 10% of the water table above the land surface will be reduced each day due to surface outflow. A detailed description of the method can be found in the following papers. We added these references in the foot notes of the table S1.**

**Zhang, Y., Li, C., Trettin, C.C., Li, H., and Sun, G., An integrated model of soil, hydrology and vegetation for carbon dynamics in wetland ecosystems. Global Biogeochemical Cycles, 16(4), 1061, 2002. doi: 10.1029/2001GB001838.**

**Zhang, Y., Li, J., Wang, X., Chen, W., Sladen, W., Dyke, L., Dredge, L., Poitevin, J., McLennan, D., Stewart, H., Kowalchuk, S., Wu, W., Kershaw, G. P., Brook, R. K. and Burn, C. R. Modelling and mapping permafrost at high spatial resolution in Wapusk National**

Park, Hudson Bay Lowlands, Can. J. Earth Sci., 49(8), 925–937, 2012. doi: 10.1139/e2012-031.

The vegetation type is listed as shrub, but in your vegetation description you state that no shrubs are present on the palsas, only on the sides. How is this choice motivated?

**[Authors' response] This choice is based on the general description of the area. Although the surface of the palsas has no shrubs the general classification for this region would be a peat bog with small-to-medium shrubs scattered across the area. The model needs to select one of the five vegetation types: coniferous forest, broad leaf forest, mixed forest, shrubs, and sedge/grass/crops. We have expanded up on this in light of these comments.**

P4, L33: Please clarify that all thermal properties and other necessary parameters and their motivations (literature, field observations) are listed in S1 and make sure that they are.

**[Authors' response] Added a column as suggested.**

P4, L30 – P5, L12: A presentation of the model discretization/mesh is lacking. It is also unclear how initial conditions were set up and how/if any spin-up procedures were performed. Was the model parameters calibrated after a spin-up from year 1900? If so, what were the initial conditions at year 1900? Was daily air temperature the only data needed for running the model for all time periods? If there is a snow wind-scouring factor, I would assume that the model also takes in snow/precipitation data.

**[Authors' response] We added a sentence about the layer divisions for this one-dimensional model. The model was initialized by running the model iteratively using the climate data in 1900 until the modelled deep ground temperature are stable. We added a sentence about that as well. The input parameters were used and kept constant for the entire simulation years, including the spin-up years. The input climate data include daily minimum and maximum air temperatures, precipitation, vapour pressure, solar radiation and wind speed. We added a sentence about the model climate input requirement and a new section in the SI about the compilation and preparation of these data, including for future scenarios.**

Please formulate this more clearly than "climatic inputs" (P4, L8).

**[Authors' response] We added a sentence about the model requirement of the inputs climate data and a new section was added in SI about the compilation of the daily data.**

P6, L10-12: I do not understand how the accuracy of the loggers and the inherent uncertainty in ERT is considered in estimating this very precise thickness value without an uncertainty range. Is this an estimate of maximum likely thickness?

**[Authors' response] Yes and we have added more information and restated a portion of the phrase. Amended the text to read:**

**Considering the accuracy of the ground temperature loggers (± 0.2°C) and uncertainty inherent to the ERT, the permafrost thickness beneath P3 was estimated to be between 4 and 8 m with our best estimate being approximately 6 m.**

P8, L30: TTOP?

**[Authors' response] Amended to:**

**Annual ground temperatures measured at or close to the base of the annual freeze-thaw layer (e.g. TTOP; see Way and Lewkowicz, 2018) at…**

P11, L4-15: So, the higher magnitude of the geothermal heat flux is used to compensate for the lack of horizontal heat fluxes in the model. How can you assume that the influence of the horizontal fluxes is stable over time, i.e. for keeping the calibrated geothermal heat flux values steady over the simulated warming periods? I understand that it is probably not possible to test this within this study, but the importance of this assumption should be noted in the text, especially as you argue for a higher geothermal heat flux in Cartwright due to smaller palsas in this location.

**[Authors' response] We agree with the above comment and have added the following statement:**

**At both sites, the geothermal fluxes were also kept constant throughout the model simulations which is a limitation of the modelling approach employed. Three-dimensional modelling would be needed to resolve the nature of expected changes in these fluxes over time as feature morphology evolves with degradation.**

P11, L9: Do you mean horizontal (instead of vertical)?

**[Authors' response] Yes. Amended accordingly.**

P12, L20: Another relevant reference about how snow influences palsa ground temperatures in Scandinavia is Sannel et al., 2016: Sannel, A.B.K., Hugelius, G., Jansson, P., Kuhry, P., 2016:

Permafrost warming in a subarctic peatland – which meteorological controls are most important? Permafrost and Periglacial Processes, doi:10.1002/ppp.1862.

**[Authors' response] Added and noted.**

P12, L26-30: Why could it not be both? The simulation tool applied here does not take into account potential feedback processes that could speed up warming/thawing with time, such as increases in lateral heat transport as permafrost bodies decrease in size, and feedback from changing topography (with melt of ground ice) to decreased wind-scouring and subsequent warming/thawing.

**[Authors' response] These feedbacks, although relevant for how permafrost may degrade in the future at these sites, are not considered to be directly attributable as being a cause of the large thermal offsets observed at these sites as referred to in these lines of text. We have added in discussion of these points elsewhere in the text however.**

P13, L8: Why does these heat flows need to be advective? See for example Kurylyk et al., 2016: Kurylyk, B. L., M. Hayashi, W. L. Quinton, J. M. McKenzie, and C. I. Voss (2016), Influence of vertical and lateral heat transfer on permafrost thaw, peatland landscape transition, and groundwater flow, Water Resour. Res., 52, 1286–1305, doi:10.1002/2015WR018057.

**[Authors' response] We have added the reference and removed advective. They could be both as noted by the reviewer.**

**Referee #2**

The manuscript describes detailed field studies of isolated bodies of permafrost along a transect in southeastern Labrador, Canada, as well as corresponding numerical simulations. I agree with reviewer #1 what the manuscript is generally well written with a clear structure and adds significant new knowledge to our understanding of isolated permafrost bodies in peatlands, but with more information needed especially on the modelling part of the study. I also recommend publication after minor revision, with the following additional comments to be addressed:

**[Authors' response] The authors' would like to thank reviewer 2 for taking the time to review this manuscript. We agree with reviewer 2's comments and have amended the manuscript accordingly.**

P1, L13: consider adding "in this region" after permafrost, as it might otherwise sound like thick peat is generally critical to permafrost.

**[Authors' response] Added.**

P1, L17: Here you mention "downscaled global warming scenarios", but there is no mentioning of downscaling in the methods section, only using multi model mean values.

**[Authors' response] The original climate model projections were regraded to common 1° by 1° global grid by Environment and Climate Change Canada and we downscaled the monthly climate scenarios to daily. This probably is not important therefore we delete this phrase.**

P3, L13-19: how does this relate to the information in P2L27, that regional air temperatures have been rapidly increasing over the last 50 years? Was the study period colder than for instance the mean of the last decade?
P3, L13-19: consider adding some more information here about the climatic conditions, like mean annual precipitation.

**[Authors' response] We have added additional discussion of climate variables and a comparison with the most recent decadal period to this section.**

P5, L6-12: which parameters were calibrated and how should be more clearly stated.

**[Authors' response] We calibrated only two parameters: snow wind-scouring factor and geothermal heat flux. We revised this the first sentence in the paragraph to clearly indicate that.**

P5, L8-9: Is this the multi-model mean from the CMIP5 archive? Were these values used directly, or just the trend? If these were used directly, how did the values correspond to the measurements in the overlap period (e.g. 2006-2016)?

**[Authors' response] We used the anomalies (difference for air temperature and relative difference for precipitation) with respect to the reference period of 1976-2005, then we derived the future monthly values based on the anomalies and the averages during the reference period at each sites. We added a new section in SI about the climate data compilation for the model.**

P6, L13-15: Here and elsewhere (e.g. P6, L30) the authors describe more (seasonal) ice than expected. Is this an indication that the study period was colder than the previous years (see comment P3, L13-19)?

**[Authors' response] We have added some additional commentary on this point.**

P7, L20: Drop "s" in "Tables 3".

**[Authors' response] Done.**

P11, L9-10: I find the explanation for the high geothermal heat fluxes needed reasonable. However, if this is really heat flow from the surroundings, is it reasonable to keep this constant throughout the simulation? Also, what is the error introduced by adding this heat at the base of a 120m soil column?

**[Authors' response] The model has only one parameter to add heat flux from the lower boundary. We calibrated it based on modelled and observed soil temperatures observed at 1m and 3 m for Cartwright site, and at 2.0m and 4.5m at Blanc Sablon site during 2014-2016 (Figure S1 and S2). It is hard to consider its variation with time using this 1D model. A two- or three-dimensional model with detailed hydrological component is needed to quantify these changes such as the studies of Kurylyk et al. (2015, doi: 10.1002/2015WR018057) and Sjöberg et al. (2016. doi:10.1002/2015WR017571). We added this limitation of the model in the discussion.**

P12, L23: TTOP should be explained here. What is it and how is it derived? If these are values derived with the TTOP model I would not call these "recorded".

**[Authors' response] At an earlier mention of TTOP we have added the definition of TTOP used in this study and have referred to Way and Lewkowicz (2018) where there is a fuller discussion of this definition and these values. We have added a reference to that study here as well.**

P15, L2-3: I would add the snow feedback to the reasons why these simulations might be too optimistic: When the PF thaws (and excess ice melts) less snow should be removed, and the wind-scouring factor should decrease, which is not accounted for here.

**[Authors' response] We have added the following discussion:**

**Degradation induced changes in palsa morphology due to melting of excess ice may further warm ground temperatures via changes in snow accumulation (increases) which are not considered in the model simulations. Correspondingly, the modelling results are likely optimistic in terms of permafrost persistence.**

P15, L22: Koven et al. (2013) does not describe regional model simulations, but global.

**[Authors' response] Agreed and amended accordingly.**

Table 3: It would be useful to have the locations in this table as well, so one would not have to go back and forth between this table and table 1. Consider adding this as an extra column here or naming the ERT profiles according to the locations (e.g. BS1, BS2, RB1, RB2 etc).

**[Authors' response] We had added a column delimiting the location for each profile.**

[revised manuscript text omitted]
. using the water-jet methodwater-jet drilling with a low horsepower pump pumping water from a nearby water body down a steel pipe to penetrating the ground. Immediately post-drilling, hBoreholes and were cased with 1-inch25 mm diameter PVC pipe immediately after drilling. The base of the permafrost was not reached at two locations (WJD01 and WJD05) due to limited water supply. Ground temperatures at four to six depths within each borehole were recorded bi-hourly with Onset Hobo V2 loggers equipped with external thermistors. Temperatures at WJD01 were measured with four high resolution ibutton loggers. Ground temperatures for four of the five boreholes are reported for the second and third full years of record to eliminate any thermal influence of water-jet drilling. At WJD05, data are provided only for the second year because this borehole was drilled one year after the others. Short data gaps in the borehole temperature records were infilled using cross-correlation between borehole depths and/or adjacent climate monitoring stations established at each location.

ERT surveys at the study sites (Table 1) were undertaken with an ABEM Terrameter LS profiling system with electrodes arranged in a Wenner configuration. ERT has been widely used for detecting permafrost bodies and estimating permafrost thickness in Canada (Briggs et al., 2016; Douglas et al., 2016; Lewkowicz et al., 2011; Minsley et al., 2016; Way and Lewkowicz, 2015), Scandinavia (Kasprzak, 2015; Sjöberg et al., 2015), the European Alps (Hauck, 2013) and the Tibet Plateau (You et al., 2013). Minimum electrode spacing was 0.5, 1 or 2 m over standard profile lengths of 40-160 m, or longer where roll-along surveys were performed, giving maximum penetration depths of approximately 6 m, 12 m or 25 m, respectively. RES2DINV software was used to invert the measured resistivities (Loke et al., 2003; Loke and Barker, 1996) with the robust inversion method. Inversion proceeded until the fifth iteration or until the RMS error dropped below 5%, whichever came first. Prior to inversion, ERT profiles were topographically corrected using a handheld GPS (Garmin Oregon 450t) to obtain start point elevations and a Brunton compass to obtain relative elevations. Modelled rTomogramsesistivities are presented as model blocks with less reliably measuredreliable blocksow (sensitivity values areas (< 0.1) faded to reflect show that these sections are less certainthe uncertainties of these sections in the resultant modelled ed 
[revised manuscript text omitted]

**Appendix S1: Compiling daily climate input data for the NEST model**

The NEST model requires continuous daily climate inputs of minimum and maximum air temperatures, precipitation, vapour pressure, solar radiation and wind speed. We compiled the input data based on climate station observations and grid datasets for the period 1900₀1-2016. Daily air temperature and precipitation observations were available during 1938-2016 for Cartwright station, and 1976-2016 for Blanc Sablon station from Environment and Climate Change Canada. McKenney and colleagues (McKenney et al., 2006; Hutchinson et al., 2009; McKenney et al., 2011) in Canadian Forest Service developed 10 km by 10 km resolution grid datasets for air temperature (minimum and maximum) and precipitation (from 1950 to 2013 for daily and 1901 to 2013 for monthly). The year 1900 was infilled by linearly extrapolating the monthly data for model initialization purpose. We downscaled the monthly data to daily from 1900 to 1949 using daily grid data from 1950 to 2013 as templates (see appendix in Zhang et al., 2012). Based on this grid data, we extended the daily climate station data to 1900.

From 2017 to 2100, we used three climate projections under RCP 2.6, 4.5 and 8.5 scenarios based on an ensemble of 29 climate models developed by the Coupled Model Intercomparison Project Phase 5 (CMIP5) (Taylor et al., 2012). The data are in common 1 degree by 1 degree global grid provided by Environment and Climate Change Canada (http://climate-scenarios.canada.ca/index.php?page=gridded-data). We first calculated anomalies (difference for air temperature and relative difference for precipitation) with respect to the reference period of 1976-2005, then we derived the future monthly values based on the anomalies and the averages during the reference period at each sites. We used the anomalies of air temperature for both the

monthly averages of daily maximum and minimum air temperatures. The monthly projections were downscaled to daily using daily grid data from 1950 to 2013 as templates.

Environment Canada and National Research Council of Canada (2007) compiled hourly climate datasets and computed solar radiation for some climate stations in Canada, including Cartwright station. We directly used these data to calculate daily vapour pressure, solar radiation and wind speed for Cartwright when data are available (1964-2005). For other periods when data were not available at Cartwright and at Blanc Sablon, we estimated daily vapour pressure and solar radiation based on the following equations (see appendix in Zhang et al., 2012)

$$V = \alpha V_{s,Tm}, \tag{1}$$

$$R = R_0 a_1 [1 - exp(-a_2 \Delta T^{a3})], \tag{2}$$

where $V$ is estimated average water vapour pressure for a day (mbar) and $V_{s,Tm}$ is saturated water vapour pressure (mbar) at the daily minimum air temperature $T_m$ (°C). $R$ is the estimated daily total solar radiation (MJ/m$^2$/d), $R_0$ is daily total solar radiation above the atmosphere (MJ/m$^2$/d), and $\Delta T$ is the diurnal temperature range (°C). $\alpha$, $a_1$, $a_2$, and $a_3$ are empirical parameters (no units) determined based on the hourly data during 1964-2015 at Cartwright station (1.05, 0.79, 0.02, 2.0, respectively). Wind speed was filled using the observations during 1954-2005 at Cartwright climate station.

**Table S1:** Parameters for ground temperature modelling with NEST at Cartwright and Blanc Sablon.

| Parameter | Cartwright (WJD02) | Blanc Sablon (WJD03) | Data sources or estimation methods |
|---|---|---|---|
| Latitude (°N) | 53.7 °N | 51.45 °N | Based on field data. |
| Peat thickness | 1.20 m | 1.75 m | Based on field data. |
| Texture of organic layers | Undecomposed peat | Undecomposed peat | Based on field data. |
| Organic matter content in mineral soils | 1.2-1.5 m: decreases from 100% to 5%, 1.5-3.2 m: 5%, then linearly decreases to 1% at 10 m | 1.75-2.0 m: decreases from 100% to 5%, 2.0-3.2 m: 5%, then linearly decreases to 1% at 10 m | Estimated based on the visual expression of the core. |
| Texture of the organic matter in mineral soils | From hemic to well decomposed at depth | From hemic to well decomposed at depth | Estimated based on the visual expression of the core. |
| Sub-peat stratigraphy |  1.2-3.2 m: silt,  3.2-10 m: sand |  1.75-3.2 m: silt,  3.2-10 m: sand | Estimated based on the cores. |
| Depth to bedrock | 10 m | 10 m | Based on field data. |
| Volumetric fraction of quartz in mineral soil | 0.1 | 0.1 | Assumed based on Dell (1959). |
| Thermal conductivity of rock | 1.5 W/(m·K) | 1.5 W/(m·K) | Based on Pollack et al. (1993)[1]. |
| Geothermal heat flux | 1.02 W/m$^2$ | 0.54 W/m$^2$ | Calibrated based on observed ground temperature. |
| Lateral surface water outflow[2] | Water table reduces 10% daily when it is above ground surface | Water table reduces 10% daily when it is above ground surface | Assumed based on local topography. |
| Lateral surface water inflow | 0 | 0 | Assumed. |
| Lateral ground water inflow and outflow | 0 | 0 | Assumed. |
| Vegetation type | Shrubs | Shrubs | Based on field data. |
| Leaf area index (peak growing season) | 0.5 | 0.2 | Based on visual expression of field conditions. |
| Snow wind-scouring factor [3] | 0.83 | 0.85 | Calibrated based on near surface soil temperature. |
| Surface albedo (no snow) | 0.1 | 0.1 | Based on Houldcroft et al. (2009) |

[1] The medium value of the observations in Canada and some surrounding sites in U.S. with observation depth < 120 m (756 sites).

[2] See Zhang et al. (2002, 2012) for detailed description.

[3] Snow wind-scouring factor is the fraction of snowfall blown away from the site.

**Table S2.** Characteristics of peatland permafrost mounds surveyed using ERT.

| ERT Profile | Mound ID | Frost table depth (cm) | Maximum mound height (m) | Inferred permafrost depth (m) | Calculated permafrost thickness (m) | Excess ice fraction[a] |
|---|---|---|---|---|---|---|
| 1 | P1 | 60 | 0.9 | 3.4 | 2.8 | 0.33 |
| 1 | P2 | 57 | 1.2 | 4.5 | 3.9 | 0.31 |
| 1 & 2 | P3 | 60 | 1.4 | 6.5 | 5.9 | 0.23 |
| 1 | P4 | 62 | 0.4 | 4.3 | 3.7 | 0.12 |
| 3 | P5 | 58 | 0.4 | 2.5 | 1.9 | 0.22 |
| 3 | P7 | 54 | 1.1 | 4.5 | 4.0 | 0.28 |
| 3 | P8 | 41 | 0.5 | 1.2 | 0.8 | 0.58 |
| 4 | P9 | 61 | 0.5 | 3.4 | 2.8 | 0.16 |
| 4 | P10 | 65 | 0.8 | 2.3 | 1.7 | 0.50 |
| 4 | P11 | 63 | 0.7 | 5.2 | 4.6 | 0.15 |
| 5 | P12 | 47 | 0.6 | 2.3 | 1.9 | 0.34 |
| 5 | P13 | 49 | 0.4 | 2.7 | 2.2 | 0.17 |
| 5 | P14 | 45 | 1.3 | 5.8 | 5.4 | 0.29 |
| 5 | P15 | 39 | 0.3 | 3.0 | 2.7 | 0.11 |
| 5 | P16 | 46 | 0.3 | 2.9 | 2.5 | 0.10 |
| 5 | P17 | 51 | 0.5 | 5.1 | 4.6 | 0.11 |
| 6 | P18 | 44 | 0.8 | 4.3 | 3.9 | 0.21 |
| 7 | P19 | 37 | 0.6 | 3.5 | 3.1 | 0.19 |
| 7 | P20 | 40 | 0.8 | 5.0 | 4.6 | 0.18 |
| 7 | P21 | 40 | 1.0 | 5.6 | 5.2 | 0.18 |
| 7 | P22 | 37 | 0.7 | 5.4 | 5.0 | 0.13 |
| 8 | P23 | 40 | 0.7 | 5.2 | 4.8 | 0.13 |
| 9 | P24 | 42 | 0.3 | 2.8 | 2.4 | 0.13 |
| 9 | P25 | 46 | 0.6 | 4.1 | 3.7 | 0.17 |
| 10 | P26 | 41 | 0.5 | 4.1 | 3.7 | 0.13 |
| 11 | P27 | 40 | 0.3 | 3.3 | 2.9 | 0.10 |

[a] Calculated using **EQ 1**.

[Figure]

**Figure S1.** Comparison of modelled and measured temperatures at different depths at the Cartwright (WJD02) borehole, 2014-2016. The thin lines are modelled and the crosses are measured values. The depths are shown in the panels. At 0.5 m, n=755; $r^2$=0.94; mean difference: -0.1°C; mean absolute difference: 0.75°C. Note: temperature scales differ among the panels.

[Figure]

**Figure S2.** Comparison of modelled and measured temperatures at different depths at the Blanc Sablon (WJD03) borehole, 2014-2016. The thin lines are modelled and the crosses are measured values. The depths are shown in the panels. At 0.25 m, n=621; $r^2$=0.94; mean difference: -0.14°C; mean absolute difference: 0.93°C. Note: temperature scales differ among the panels.

[Figure]

**Figure S3.** Monthly ground temperatures (n=82; black open circles) measured at 50 cm depth (1991-2012) on a palsa surfacein a palsa field, about 6 km north of near Blanc Sablon site (from Allard et al. 2014).

[Figure]

**Figure S4.** Centred 15-year average apparent thermal offsets for historical and future climate scenarios at Blanc Sablon, QC. Thicker lines denote the presence of permafrost and thinner lines denote the absence of permafrost.

---

## Author Response (AR2)

**Editor Decision: Publish subject to minor revisions (review by editor)** (23 Apr 2018) by Christian Hauck
Comments to the Author:
Dear authors,

thank you very much for your revised manuscript and your author's comments, which respond very well to the points and queries raised by the two reviewers. I re-read the manuscript myself, and have some minor comments/questions to add, which are detailed in the following. I would like to ask you to respond to them point-by-point, and revise your manuscript accordingly. Hereafter, the manuscript can be accepted for publication in The Cryosphere.

Kind regards,
Christian Hauck
Editor

**[Authors' response] We agree with most of the commentary and have responded accordingly. Likewise, we have amended the manuscript in accordance with these suggestions.**

*Uncertainties associated with ERT and permafrost thickness measurements.*
Detailed comments:
p4, line 25-26: "sensitivity values < 0.1". This point was raised by reviewer 1 ("what is meant by low sensitivity areas and how does it reflect uncertainties ?"), but was not really answered in the author's response. Where does this sensitivity value come from and how is it calculated ? Why do you use a limit of 0.1 ? This has to be justified and the explanation also given in the paper (or referenced).

**[Authors' response] As model blocks are more effective at showing contrast, we decided to use this approach but using the RES2DINV software it is necessary to use a means (such as sensitivity values) to limit the area considered for analyses. Model block sensitivity values are a measure of the amount of information about the resistivity contained in the measured block and are discussed in detail in the RES2DINV software manual that was produced based on Loke and Barker (1996) and Loke et al (2003). Higher sensitivity values correspond to more reliable model resistivity values, as such using sensitivities for limiting the area visualized. The selection of values less than 0.1 was on the basis of manual interpretation of the profiles which led to the selection of 0.1 given that values below that point were consistently representing areas that were at the deepest portions of the profiles.**

p6, line 9-16 (and in the following analysis of ERT results and EIC calculations):
The uncertainty analysis of the ERT/T-based permafrost thicknesses is only given for P3. Why not for all examples in this paragraph and even more, all values/data in Table 3 ? Without additional information, these highly accurate estimations

Again, this was mentioned also by reviewer 1 ("P6, L10-12: I do not understand how the accuracy of the loggers and the inherent uncertainty in ERT is considered in estimating this very precise thickness value without an uncertainty range. Is this an estimate of maximum likely thickness?").

Are such accurate numbers (e.g. 5.4m - p8, l28) drawn from the rather coarse ERT results alone ? Not to speak of the inherent uncertainties of the inversion process...maybe using thicknesses such as 5, 5.5m etc would be more appropriate, and/or the use of uncertainty ranges.

This uncertainty would propagate also to the EIC values given later in the manuscript.

Table 3: Inferred maximum permafrost thickness: in a similar way as reviewer 1, I do not understand how these very accurate thicknesses were derived from ERT without giving an uncertainty range ? See my comments above.

**[Authors' response] We have amended our approach for calculating uncertainties associated with the estimation of permafrost thicknesses. We agree that we have provided too high of a precision. As such, we conducted some additional analysis and have settled on a permafrost thickness error of ±0.5 m as being reasonable as this would constitute approximately one model block for most profiles. As such, we have provided ranges for values provided throughout the text, manuscript SI, tables and excess ice fraction figure. Where ground temperatures have been used to estimate permafrost thicknesses in conjunction with the ERT data, we have highlighted this.**

**However, to use ERT alone to help delineate the error associated with the direct evaluation of permafrost thicknesses, we have generated a logistical regression curve which estimates the probability that ground is frozen for a given modelled resistivity value (Ω.m).**

**First, we matched the xyz block resistivities observed at the near-surface layer (or layer closest to the depth of frozen ground – typically 0.5 m block) to the frost table measurements taken along the ERT profiles. Where frozen ground was determined as present in the upper 1 m, we coded the location as 1 and where it was absent we coded it at 0. Data from all the sites were then pooled and a logistic regression analysis was performed enabling generation of a probability of frozen ground curve for modelled resistivity values (see figure below).**

**This approach assumes that the substrate of the upper layers is representative to those deeper down which is not necessarily applicable in all cases but for the purposes of these thin permafrost features it should be reasonably representative. Permafrost probabilities for each site were then estimated from the ERT block data using the logistic regression fit and the modelled apparent resistivities (see figures below as an example). This analyses allowed us to examine what the typical level of ambiguity was for permafrost determination at the base of palsas and led us to the conclusion that it was reasonable to assume an uncertainty of 1 block.**

[Figure]

[Figure]

[Figure]

[Figure]

[Figure]

p4, line 33ff:
This is not clear to me: do you mean "divided into layers of 0.1 m thickness within the uppermost meter" ?

**[Authors' response] Amended for clarity.**

p5, line 4 (and at many places in the manuscript: I think the term"gridded dataset" is more common, at least in the climate community.
I would suggest using "gridded dataset" instead of "grid dataset" throughout the manuscript and in the supplementary material text.

**[Authors' response] Agreed and amended.**

p15, l30: "global climate model predict"

**[Authors' response] Amended to:**

**"as predicted by global climate models"**

p16, line 13: "as well as possible heat transport from sub-permafrost groundwater" - this has strictly speaking not been mentioned before. If it appears in the conclusion, it should also appear in the results or discussion section.

**[Authors' response] We had added a mention of this to the discussion. See amended manuscript.**

**"heat transport from sub-permafrost groundwater"**

Supplementary material:
Appendix S1, first paragraph, last sentence: "the daily climate station data backwards to 1900"

**[Authors' response] Amended to:**

**"The year 1900 was infilled by linearly extrapolating the monthly data for model initialization purpose."**

S1, second paragraph: "based on the anomalies and the averages during the reference period at each sites" --> "site" instead of "sites"

**[Authors' response] Amended.**

S1, general: This seems to me a very basic downscaling approach (delta approach, if I understood it right), which does neither take into account local conditions (or were the 1° climate data somehow downscaled to local conditions via meteo stations ?) nor the possibility that minimum and maximum temperature trends will develop differently in a future climate.

Do you have any justification/prior studies which use this approach ?

Otherwise, in my opinion, the scenarios used can be seen rather as a sensitivity study (what would happen under different climate conditions) than a real projection into the future. If this is the case, then it should be stated accordingly in the manuscript.

**[Authors' response]**

**The climate record of the coastal Labrador region is highly variable and its future evolution is uncertain due to broad-scale uncertainties associated with ocean-atmospheric teleconnections and other factors (Brown et al., 2012; Way and Viau, 2015; Grenier et al., 2015). The authors have made the choice to use a simple downscaling method (delta) instead of higher complexity approaches because in coastal Labrador there is considerable potential for introducing additional sources of error through ocean-atmospheric-cryosphere feedbacks and other interactions that are not consistently represented by GCMs (Grenier et al., 2015; Ekstrom et al., 2015). Way and Viau (2015) showed that over the historical record air temperatures followed the CMIP5 multi-model mean with a higher fidelity than for several other widely-used climate models. As such, we feel that the best estimate of the climate for the periods analyzed will follow this overall trend. We do acknowledge that the evolution of Tmin and Tmax and year-to-year variability may impact the potential evolution of permafrost but we also disagree that using non-GCM downscaled data is equivalent to a sensitivity study. Further, there is not a consistent set of evidence that more complicated downscaling methods are inherently more accurate relative to delta approaches for simple variables like air temperature (Ekstrom et al., 2015). We agree that the scenarios differ with the climate models and their new development and configurations. We added a sentence to show the changes of projected air temperatures to the main manuscript and have added more information on our downscaling method to the SI.**

**Added to the methods:**

**"Under the scenarios of RCP 2.6, 4.5 and 8.5, air temperature was projected to increase 1.0, 2.3, and 5.5ºC from the 2010s to the 2019s, respectively."**

S1, last sentence: "wind speed data was filled..."

**[Authors' response] Amended to:**

**"Wind speed during 1954-2005 are from observations at Cartwright climate station. In other periods, we directly used the observed wind speed during 1954-2005 to fill the data gaps."**

---

## Author Response (AR3)

**Authors' response to editor**

*[editor comment]: thank you very much for your revised manuscript and author reply. I found your answers to my comments very satisfactory, but would suggest to include them (or more parts of them) also in the paper, see my comments in the annotated manuscript. Apart from that (and some very small typos), the manuscript is ready for publication in The Cryosphere from my point of view.*

*Thank you very much again for submitting your manuscript to TC,*
*kind regards,*

*Christian Hauck*
*Editor*

**[Authors' response to editor]: We would like to thank the editor for these useful comments. we have amended the text and SI in accordance with the editor's comments. Thank you.**

*[editor comment]: I would suggest subheaders for the three different methodological parts: climate data and boreholes; ERT; thermal modelling*

**[Authors' response to editor]: Amended accordingly.**

*[editor comment]: the explanation given in the author response should also be incorporated here. It may also suffice to mention that with "sensitivity values" you mean the Res2dinv-calculated sensitivity values. This might be understood from the reference given, but I guess not for the general reader.*

**[Authors' response to editor]: We agree with this comment and have made this portion more clear in terms of the originator of the sensitivity calculation method.**

*[editor comment]: similar to comment above: your explanation given in the last author comment was more informative and should be included here. Of course not all of it, but enough to understand why you may assume an error of +-0.5m uncertainty. It might also be an option to include this explanation in the supplementary material and reference it here.*

**[Authors' response to editor]: After some discussions, we have decided to make a short mention of it here and link to a new appendix (S1) that has been created with two**

**supplemental figures for the purposes of discussing the method used for estimating the uncertainty bounds. We hope this helps elucidate the process and approach used for the readership. We hope that standardized approaches are able to be developed to ensure consistent practices across study areas in the future.**

*[editor comment]: should be chapter 4*

**[Authors' response to editor]: Amended accordingly**

*[editor comment]: geothermal heat fluxes*

**[Authors' response to editor]: Amended accordingly**

[revised manuscript text omitted]

**Appendix S1: Compiling daily climate input data for the NEST model**

The NEST model requires continuous daily climate inputs of minimum and maximum air temperatures, precipitation, vapour pressure, solar radiation and wind speed. We compiled the input data based on climate station observations and gridded datasets for the period 1901-2016. Daily air temperature and precipitation observations were available during 1938-2016 for Cartwright station, and 1976-2016 for Blanc-Sablon station from Environment and Climate Change Canada. McKenney and colleagues (McKenney et al., 2006; Hutchinson et al., 2009; McKenney et al., 2011) in Canadian Forest Service developed 10 km by 10 km resolution gridded datasets for air temperature (minimum and maximum) and precipitation (from 1950 to 2013 for daily and 1901 to 2013 for monthly). The year 1900 was infilled by linearly extrapolating the monthly data for model initialization purpose. We downscaled the monthly data to daily from 1900 to 1949 using daily gridded data from 1950 to 2013 as templates (see appendix in Zhang et al., 2012). Based on this gridded data, we extended the daily climate station data to 1900.

From 2017 to 2100, we used three climate projections under RCP 2.6, 4.5 and 8.5 scenarios based on an ensemble of 29 climate models developed by the Coupled Model Intercomparison Project Phase 5 (CMIP5) (Taylor et al., 2012). The data are in common 1 degree by 1 degree global grid provided by Environment and Climate Change Canada (http://climate-scenarios.canada.ca/index.php?page=gridded-data). The climate record of the coastal Labrador region is considered to be highly variable and its future evolution is uncertain due to broad-scale uncertainties associated with ocean-atmospheric teleconnections and other factors (Brown et al., 2012; Way and Viau, 2015; Grenier et al., 2015). The authors have made the choice to use a simple downscaling method (delta) instead of higher complexity approaches because in coastal Labrador there is considerable potential for introducing additional sources of error through ocean-atmosphere-cryosphere feedbacks and other interactions that are not consistently represented by GCMs (Grenier et al., 2015; Ekström et al., 2015). We do acknowledge that the evolution of Tmin and Tmax and year-to-year variability may impact the potential evolution of permafrost. However, there is not a consistent set of evidence that more complicated downscaling methods are inherently more accurate relative to delta approaches for simple variables like air temperature (Ekström et al., 2015). We first calculated anomalies (difference for air temperature and relative difference for precipitation) with respect to the reference period of 1976-2005, then we derived the future monthly values based on the anomalies and the averages during the reference period at each site. We used the anomalies of air temperature for both the monthly averages of daily maximum and minimum air

temperatures. The monthly projections were downscaled to daily using daily gridded data from 1950 to 2013 as templates.

Environment Canada and National Research Council of Canada (2007) compiled hourly climate datasets and computed solar radiation for some climate stations in Canada, including Cartwright station. We directly used these data to calculate daily vapour pressure, solar radiation and wind speed for Cartwright when data are available (1964-2005). For other periods when data were not available at Cartwright and at Blanc Sablon, we estimated daily vapour pressure and solar radiation based on the following equations (see appendix in Zhang et al., 2012)

$$V = \alpha V_{s,Tm},$$ (1)

$$R = R_0 a_1 [1 - exp(-a_2 \Delta T^{a3})],$$ (2)

where $V$ is estimated average water vapour pressure for a day (mbar) and $V_{s,Tm}$ is saturated water vapour pressure (mbar) at the daily minimum air temperature $T_m$ (°C). $R$ is the estimated daily total solar radiation (MJ/m²/d), $R_0$ is daily total solar radiation above the atmosphere (MJ/m²/d), and $\Delta T$ is the diurnal temperature range (°C). $\alpha$, $a_1$, $a_2$, and $a_3$ are empirical parameters (no units) determined based on the hourly data during 1964-2015 at Cartwright station (1.05, 0.79, 0.02, 2.0, respectively). Wind speed during 1954-2005 are from observations at Cartwright climate station. In other periods, we directly used the observed wind speed during 1954-2005 to fill the data gaps.

**Appendix S1: Uncertainties in permafrost thickness estimation and presentation with ERT**

ERT profiles are presented as model blocks because they are effective at showing contrast with depth. Profiles were faded in areas with lower calculated sensitivity values (<0.1; RES2DINV method) to limit the area considered during analysis. Model block sensitivity values are a measure of the amount of information about the resistivity contained in the measured block and are discussed in detail in the RES2DINV software manual that was produced based on Loke and Barker (1996) and Loke et al (2003). Higher sensitivity values correspond to more reliable model resistivity values. The selection of values less than 0.1 was on the basis of manual interpretation of the profiles.

In order to estimate permafrost thicknesses from ERT alone we combined frost table probing and modelled resistivities for each of the ERT profiles. First, we matched the xyz block resistivities at a depth of 0.5-1.0 m to the frost table measurements taken along the ERT profiles. Where a frost table was recorded, the apparent resistivity in the layer from 0.5-1.0 m was linked to frozen ground presence (value of 1 in the subsequent logistic regression). Where no frost table was recorded within the top 120 cm (maximum probing depth), the apparent resistivity for 0.5-1.0 m was associated with unfrozen soil (value of 0 in the logistic regression). Data from all sites were pooled for the logistic regression analysis (Fig. S1). The best-fit curve was then used with the ERT modelled apparent resistivity blocks to generate permafrost probabilities (Fig. S2). This

approach assumes that the substrate in the upper layers is representative of those at depth, which may not always be correct. Permafrost probability typically dropped very sharply with depth, from >90% to <10% between two vertically adjacent model blocks. At some locations, a single block with intermediate probability existed between these two extremes. Based on these results, in the absence of ground temperatures measured in boreholes, we adopted a permafrost thickness error of ±0.5 m, which is equivalent to ±one model block for most profiles.

**Appendix S2: Compiling daily climate input data for the NEST model**

The NEST model requires continuous daily climate inputs of minimum and maximum air temperatures, precipitation, vapour pressure, solar radiation and wind speed. We compiled the input data based on climate station observations and gridded datasets for the period 1901-2016. Daily air temperature and precipitation observations were available during 1938-2016 for Cartwright station, and 1976-2016 for Blanc Sablon station from Environment and Climate Change Canada. McKenney and colleagues (McKenney et al., 2006; Hutchinson et al., 2009; McKenney et al., 2011) in Canadian Forest Service developed 10 km by 10 km resolution gridded datasets for air temperature (minimum and maximum) and precipitation (from 1950 to 2013 for daily and 1901 to 2013 for monthly). The year 1900 was infilled by linearly extrapolating the monthly data for model initialization purpose. We downscaled the monthly data to daily from 1900 to 1949 using daily gridded data from 1950 to 2013 as templates (see appendix in Zhang et al., 2012). Based on this gridded data, we extended the daily climate station data to 1900.

From 2017 to 2100, we used three climate projections under RCP 2.6, 4.5 and 8.5 scenarios based on an ensemble of 29 climate models developed by the Coupled Model Intercomparison Project Phase 5 (CMIP5) (Taylor et al., 2012). The data are in common 1 degree by 1 degree global grid provided by Environment and Climate Change Canada (http://climate-scenarios.canada.ca/index.php?page=gridded-data). The climate record of the coastal Labrador region is considered to be highly variable and its future evolution is uncertain due to broad-scale uncertainties associated with ocean-atmospheric teleconnections and other factors (Brown et al., 2012; Way and Viau, 2015; Grenier et al., 2015). The authors have made the choice to use a

simple downscaling method (delta) instead of higher complexity approaches because in coastal Labrador there is considerable potential for introducing additional sources of error through ocean-atmospheric-cryosphere feedbacks and other interactions that are not consistently represented by GCMs (Grenier et al., 2015; Ekström et al., 2015). We do acknowledge that the evolution of Tmin and Tmax and year-to-year variability may impact the potential evolution of permafrost. However, there is not a consistent set of evidence that more complicated downscaling methods are inherently more accurate relative to delta approaches for simple variables like air temperature (Ekström et al., 2015). We first calculated anomalies (difference for air temperature and relative difference for precipitation) with respect to the reference period of 1976-2005, then we derived the future monthly values based on the anomalies and the averages during the reference period at each site. We used the anomalies of air temperature for both the monthly averages of daily maximum and minimum air temperatures. The monthly projections were downscaled to daily using daily gridded data from 1950 to 2013 as templates. Environment Canada and National Research Council of Canada (2007) compiled hourly climate datasets and computed solar radiation for some climate stations in Canada, including Cartwright station. We directly used these data to calculate daily vapour pressure, solar radiation and wind speed for Cartwright when data are available (1964-2005). For other periods when data were not available at Cartwright and at Blanc Sablon, we estimated daily vapour pressure and solar radiation based on the following equations (see appendix in Zhang et al., 2012)

$$V = \alpha V_{s,Tm}, \tag{1}$$

$$R = R_0 a_1 [1 - exp(-a_2 \Delta T^{a3})], \tag{2}$$

where $V$ is estimated average water vapour pressure for a day (mbar) and $V_{s,Tm}$ is saturated water vapour pressure (mbar) at the daily minimum air temperature $T_m$ (°C). $R$ is the estimated daily total solar radiation (MJ/m$^2$/d), $R_0$ is daily total solar radiation above the atmosphere (MJ/m$^2$/d), and $\Delta T$ is the diurnal temperature range (°C). $\alpha$, $a_1$, $a_2$, and $a_3$ are empirical parameters (no units) determined based on the hourly data during 1964-2015 at Cartwright station (1.05, 0.79, 0.02, 2.0, respectively). Wind speed during 1954-2005 are from observations at Cartwright climate station. In other periods, we directly used the observed wind speed during 1954-2005 to fill the data gaps.

**Table S1:** Parameters for ground temperature modelling with NEST at Cartwright and Blanc Sablon.

| Parameter | Cartwright (WJD02) | Blanc Sablon (WJD03) | Data sources or estimation methods |
|---|---|---|---|
| Latitude (°N) | 53.7 °N | 51.45 °N | Based on field data. |
| Peat thickness | 1.20 m | 1.75 m | Based on field data. |
| Texture of organic layers | Undecomposed peat | Undecomposed peat | Based on field data. |
| Organic matter content in mineral soils | 1.2-1.5 m: decreases from 100% to 5%, 1.5-3.2 m: 5%, then linearly decreases to 1% at 10 m | 1.75-2.0 m: decreases from 100% to 5%, 2.0-3.2 m: 5%, then linearly decreases to 1% at 10 m | Estimated based on the visual expression of the core. |
| Texture of the organic matter in mineral soils | From hemic to well decomposed at depth | From hemic to well decomposed at depth | Estimated based on the visual expression of the core. |
| Sub-peat stratigraphy | 1.2-3.2 m: silt, 3.2-10 m: sand | 1.75-3.2 m: silt, 3.2-10 m: sand | Estimated based on the cores. |
| Depth to bedrock | 10 m | 10 m | Based on field data. |
| Volumetric fraction of quartz in mineral soil | 0.1 | 0.1 | Assumed based on Dell (1959). |
| Thermal conductivity of rock | 1.5 W/(m·K) | 1.5 W/(m·K) | Based on Pollack et al. (1993)[1]. |
| Geothermal heat flux | 1.02 W/m$^2$ | 0.54 W/m$^2$ | Calibrated based on observed ground temperature. |
| Lateral surface water outflow[2] | Water table reduces 10% daily when it is above ground surface | Water table reduces 10% daily when it is above ground surface | Assumed based on local topography. |
| Lateral surface water inflow | 0 | 0 | Assumed. |
| Lateral ground water inflow and outflow | 0 | 0 | Assumed. |
| Vegetation type | Shrubs | Shrubs | Based on field data. |
| Leaf area index (peak growing season) | 0.5 | 0.2 | Based on visual expression of field conditions. |
| Snow wind-scouring factor [3] | 0.83 | 0.85 | Calibrated based on near surface soil temperature. |
| Surface albedo (no snow) | 0.1 | 0.1 | Based on Houldcroft et al. (2009) |

[1] The medium value of the observations in Canada and some surrounding sites in U.S. with observation depth < 120 m (756 sites).
[2] See Zhang et al. (2002, 2012) for detailed description.
[3] Snow wind-scouring factor is the fraction of snowfall blown away from the site.

**Table S2.** Characteristics of peatland permafrost mounds surveyed using ERT.

| ERT Profile | Mound ID | Frost table depth (cm) | Maximum mound height (m) | Inferred permafrost depth (m) | Calculated permafrost thickness (m) | Excess ice fraction[a] |
|---|---|---|---|---|---|---|
| 1 | P1 | 60 | 0.9 | 3.4 ± 0.5 | 2.3 – 3.3 | 0.28 – 0.40 |
| 1 | P2 | 57 | 1.2 | 4.5 ± 0.5 | 3.4 – 4.4 | 0.28 – 0.36 |
| 1 & 2 | P3 | 60 | 1.4 | 6.5 ± 0.5 | 5.4 – 6.4 | 0.22 – 0.26 |
| 1 | P4 | 62 | 0.4 | 4.3 ± 0.5 | 3.2 – 4.2 | 0.11 – 0.14 |
| 3 | P5 | 58 | 0.4 | 2.5 ± 0.5 | 1.4 – 2.4 | 0.17 – 0.30 |
| 3 | P7 | 54 | 1.1 | 4.5 ± 0.5 | 3.5 – 4.5 | 0.24 – 0.32 |
| 3 | P8 | 41 | 0.5 | 1.2 ± 0.5 | 0.3 – 1.3 | 0.36 – 1.59 |
| 4 | P9 | 61 | 0.5 | 3.4 ± 0.5 | 2.3 – 3.3 | 0.14 – 0.20 |
| 4 | P10 | 65 | 0.8 | 2.3 ± 0.5 | 1.2 – 2.2 | 0.38 – 0.71 |
| 4 | P11 | 63 | 0.7 | 5.2 ± 0.5 | 4.1 – 5.1 | 0.13 – 0.16 |
| 5 | P12 | 47 | 0.6 | 2.3 ± 0.5 | 1.4 – 2.4 | 0.27 – 0.46 |
| 5 | P13 | 49 | 0.4 | 2.7 ± 0.5 | 1.7 – 2.7 | 0.15 – 0.22 |
| 5 | P14 | 45 | 1.3 | 5.8 ± 0.5 | 4.9 – 5.9 | 0.22 – 0.27 |
| 5 | P15 | 39 | 0.3 | 3.0 ± 0.5 | 2.2 – 3.2 | 0.09 – 0.13 |
| 5 | P16 | 46 | 0.3 | 2.9 ± 0.5 | 2.0 – 3.0 | 0.08 – 0.13 |
| 5 | P17 | 51 | 0.5 | 5.1 ± 0.5 | 4.1 – 5.1 | 0.10 – 0.13 |
| 6 | P18 | 44 | 0.8 | 4.3 ± 0.5 | 3.4 – 4.4 | 0.19 – 0.24 |
| 7 | P19 | 37 | 0.6 | 3.5 ± 0.5 | 2.6 – 3.6 | 0.16 – 0.23 |
| 7 | P20 | 40 | 0.8 | 5.0 ± 0.5 | 4.1 – 5.1 | 0.16 – 0.20 |
| 7 | P21 | 40 | 1.0 | 5.6 ± 0.5 | 4.7 – 5.7 | 0.17 – 0.20 |
| 7 | P22 | 37 | 0.7 | 5.4 ± 0.5 | 4.5 – 5.5 | 0.12 – 0.14 |
| 8 | P23 | 40 | 0.7 | 5.2 ± 0.5 | 4.3 – 5.3 | 0.12 – 0.15 |
| 9 | P24 | 42 | 0.3 | 2.8 ± 0.5 | 1.9 – 2.9 | 0.11 – 0.17 |
| 9 | P25 | 46 | 0.6 | 4.1 ± 0.5 | 3.2 – 4.2 | 0.15 – 0.19 |
| 10 | P26 | 41 | 0.5 | 4.1 ± 0.5 | 3.2 – 4.2 | 0.11 – 0.15 |
| 11 | P27 | 40 | 0.3 | 3.3 ± 0.5 | 2.4 – 3.4 | 0.08 – 0.12 |

[a] Calculated using **EQ 1**.

[Figure]

**Figure S1:** Logistic regression curve (AIC: 358.01; pseudo $r^2$: 0.4) generated from co-located observations of frozen ground (thaw depth probing) and near-surface modelled resistivities extracted from model blocks (typically 0.5-1.0 m depths).

[Figure]

**Figure S2:** ERT profile #7 from Cartwright, NL. (a) Profile depicting the modelled resistivities; (b) Profile of predicted permafrost probabilities derived using the logistic regression curve presented in S1 with the data presented in (a). Hatching shows standard plot depth of investigation presented in RES2DINV. Plots of profiles were generated in R v3.3 using a custom script.

[Figure]

**Figure S3.** Comparison of modelled and measured temperatures at different depths at the Cartwright (WJD02) borehole, 2014-2016. The thin lines are modelled and the crosses are measured values. The depths are shown in the panels. At 0.5 m, n=755; $r^2$=0.94; mean difference: -0.1°C; mean absolute difference: 0.75°C. Note: temperature scales differ among the panels.

[Figure]

**Figure S42.** Comparison of modelled and measured temperatures at different depths at the Blanc Sablon (WJD03) borehole, 2014-2016. The thin lines are modelled and the crosses are measured values. The depths are shown in the panels. At 0.25 m, n=621; $r^2$=0.94; mean difference: -0.14°C; mean absolute difference: 0.93°C. Note: temperature scales differ among the panels.

[Figure]

**Figure S53.** Monthly ground temperatures (n=82; black open circles) measured at 50 cm depth (1991-2012) on a palsa surface, about 6 km north of Blanc Sablon site (from Allard et al. 2014).

[Figure]

**Figure S64.** Centred 15-year average apparent thermal offsets for historical and future climate scenarios at Blanc Sablon, QC. Thicker lines denote the presence of permafrost and thinner lines denote the absence of permafrost.